# Induction Meets Biology:
# Mechanisms of Repeat Detection in Protein Language Models

**Gal Pomerants** [1]   **Yaniv Nikankin** [1]   **Anja Reusch** [1]   **Tomer Tsaban** [2]   **Ora Schueler-Furman** [2]   **Yonatan Belinkov** [1 3]

## Abstract

Protein sequences are abundant in repeating segments, both as exact copies and as approximate segments with mutations. These repeats are important for protein structure and function, motivating decades of algorithmic work on repeat identification. Recent work has shown that protein language models (PLMs) identify repeats, by examining their behavior in masked-token prediction. To elucidate their internal mechanisms, we investigate how PLMs detect both exact and approximate repeats. We find that the mechanism for approximate repeats functionally subsumes that of exact repeats. We then characterize this mechanism, revealing two main stages: PLMs first build feature representations using both general positional attention heads and biologically specialized components, such as neurons that encode amino-acid similarity. Then, induction heads attend to aligned tokens across repeated segments, promoting the correct answer. Our results reveal how PLMs solve this biological task by combining language-based pattern matching with specialized biological knowledge, thereby establishing a basis for studying more complex evolutionary processes in PLMs.[1]

## 1 Introduction

Protein sequences frequently contain internal repeats—amino acid segments that recur multiple times within the same protein (Delucchi et al., 2020; Marcotte et al., 1999)—and are critical for important biological properties, such

as structural stability and molecular interactions (Andrade et al., 2001). These repeating sequences are not necessarily identical: evolutionary events cause originally identical segments to diverge, accumulating mutations that result in approximate repeats sharing biochemical similarity rather than exact sequence identity (see Appendix A for further background on the evolution of protein sequence repeats). These variations make repeat identification a challenging task (Luo & Nijveen, 2014), for which numerous algorithms were proposed, including both sequence-based methods (Heger & Holm, 2000; Jorda & Kajava, 2009; Newman & Cooper, 2007; Biegert & Söding, 2008; Szklarczyk & Heringa, 2004) and structure-based approaches (Hrabe & Godzik, 2014; Bliven et al., 2019; Kim et al., 2010; Mozaffari et al., 2024; Hirsh et al., 2016).

Recently, protein language models (PLMs) have demonstrated remarkable ability to recognize repeated patterns when performing masked token prediction (Kantroo et al., 2025), but *how* they do so has not been studied before. Understanding their internal mechanisms could inform improved repeat-detection pipelines and increase confidence in PLM predictions (Hu et al., 2022).

To address this gap, we reverse-engineer the internal mechanisms enabling PLMs to recognize and complete repeating sequences. We identify a circuit—a subset of interconnected model components (Elhage et al., 2021)—that faithfully captures the model's behavior in predicting masked tokens within both identical and approximate repeats. We find that circuits for identical and approximate repeats largely overlap, with the approximate-repeat circuit functionally generalizing the identical-repeat circuit.

Zooming into individual circuit components—attention heads and multi-layer perceptron (MLP) neurons—reveals two high-level categories. First, some exhibit repetition-related functionalities analogous to language models: induction-like attention heads attending to aligned repeat tokens (Olsson et al., 2022), heads attending to fixed positional offsets (Elhage et al., 2021), and neurons activating within repetitions (Hiraoka & Inui, 2025) (Fig. 1 left). Second are biologically specialized components: attention heads biased toward specific amino acids and neurons responding to individual amino acids or to biochemically similar groups

---

[1]Technion - Israel Institute of Technology [2]The Hebrew University of Jerusalem, Israel [3]Kempner Institute, Harvard University. Correspondence to: Gal Pomerants <galkesten@campus.technion.ac.il, galkesten@gmail.com>.

*Proceedings of the 43rd International Conference on Machine Learning*, Seoul, South Korea. PMLR 306, 2026. Copyright 2026 by the author(s).

[1]Code and data available at https://github.com/technion-cs-nlp/plms-repeats-circuits.

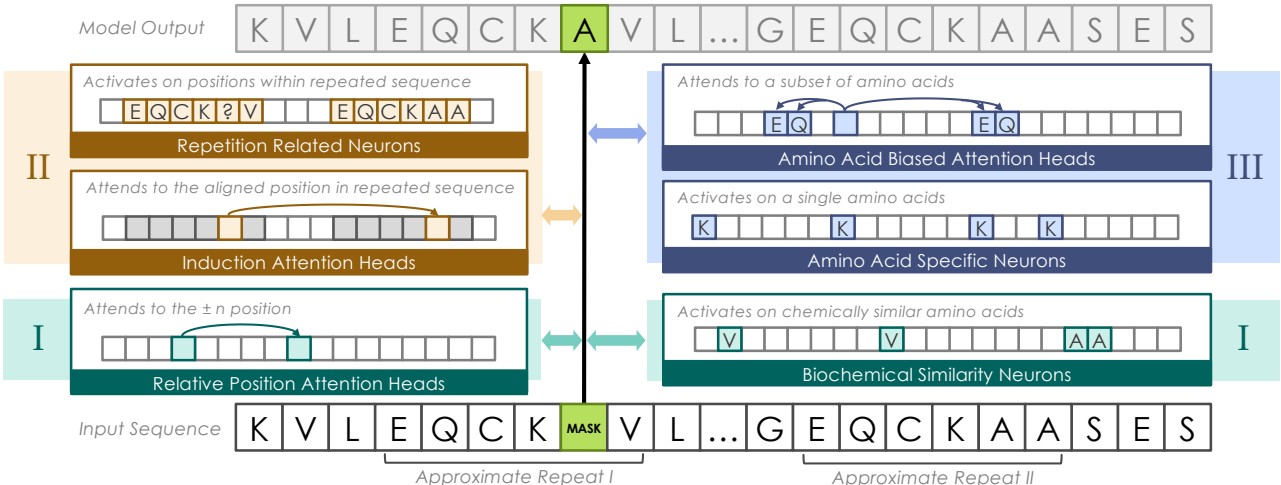

*Figure 1.* **Visualization of the repeat identification mechanism.** The model predicts the masked token by integrating repetition-related (left) and biological features (right). (I) First, relative-position attention heads attend to tokens located at fixed offsets ($\pm n$) from the masked position, followed by the activation of biologically specialized neurons, such as neurons that selective for biochemically similar amino acids. (II) In middle layers, induction heads attend to the token aligned with the masked position in the other repeat instance and copy its information, enabling retrieval of the correct amino acid, while repetition neurons play an inhibitory role. (III) Finally, MLP neurons refine the final masked token distribution, with amino-acid-biased attention heads also contributing to the prediction.

with high substitution likelihood (Fig. 1 right).

We analyze how these components combine to identify and complete repeats, uncovering a three-phase mechanism: (I) In early layers, relative-position heads and biochemical neurons establish context within repeats; (II) in middle layers, induction heads activate on aligned repeat tokens and drive prediction, while repetition-sensitive neurons play an inhibitory role; and (III) in the final layers, MLP neurons play the primary role in refining the output distribution, with amino-acid–biased attention heads contributing a smaller, supplementary effect to the final prediction.

Finally, we compare two prominent PLMs—ESM-3 (Hayes et al., 2025) and ESM-C (ESM Team, 2024)—finding they share a similar mechanism, with one crucial difference: ESM-3 engages neurons sensitive to protein secondary structure for repeat identification, while ESM-C does not. We relate this to differences in their training regimes and data.

Our work is the first to characterize repeat identification in PLMs, revealing how general induction mechanisms integrate protein-specific components encoding biological features. We reveal that many model components are directly interpretable, and by reverse-engineering circuits for both exact and approximate repeat completion, we deepen our understanding of how PLMs utilize evolutionary patterns.

## 2 Experimental Settings

**Repeat identification as masked language modeling.** To study the mechanisms by which PLMs identify and complete repeating sequences, we formulate repeat identification

in protein sequences as a masked language modeling task (Kantroo et al., 2025). The model receives as input a protein sequence tokenized into individual amino acid tokens. The input sequence contains a repeated segment that appears exactly twice; these repeat occurrences may be identical or may differ by a small number of amino acid substitutions.[2] We mask a single amino acid token in one of the repeat occurrences, and the model predicts the masked token using the remaining context, including information from the other repeat occurrence. An example is shown in Fig. 1.

To understand the mechanisms PLMs employ for both general and biologically-plausible repeating sequences, we design three task variants. First, we focus on the general case of repeating sequences by using *synthetic* sequences. Each synthetic sequence consists of 200 random amino acid tokens, containing two *identical* copies of a sampled segment with length between 10 and 30. Second, to examine whether the repeat-completion mechanism generalizes to natural proteins, we use naturally occurring protein sequences with two *identical* repeat occurrences. We curate these proteins from UniRef50 (Suzek et al., 2014) and identify repeats using the RADAR algorithm (Heger & Holm, 2000). Third, to test robustness to naturally-occurring variations, we examine *approximate* repeats. We curate naturally occurring proteins where the two repeating occurrences differ by substitutions. We restrict the substitution rate to at most 50%. In this setting, we mask positions that are identical to their aligned

---

[2]We additionally considered repeats with insertion and deletion mutations; however, model performance in this setting was low and unstable. See Appendix C for details.

counterparts but are adjacent to a substitution site, ensuring that local sequence similarity is disrupted while the repeat relationship remains detectable. Together, these variants allow us to assess whether repeat-completion mechanisms are universal or specialized, and whether they depend on exact sequence matching or can accommodate variation. Full details of the data curation process are described in Appendix B, with example inputs shown in Table 3.

**Models.** We analyze two PLMs: ESM-3-open-1.4B (Hayes et al., 2025) and ESM-C-600M (ESM Team, 2024). Both are transformer encoders trained with masked language modeling at the amino acid token level. ESM-3 is trained on multiple modalities (sequence, structure, and functional annotations), while ESM-C uses sequence inputs only; for ESM-3 we use only the sequence track for input and prediction. We evaluate both models on all three task variants (Table 1), finding they perform above 99% on synthetic and identical repeats, and 79% on approximate ones. (See Appendix D for evaluation details and Appendix E for an analysis of failure cases in the approximate-repeat setting.) All further analyses are restricted to successful instances. The main paper focuses on ESM-3, with ESM-C results in Appendix F.

*Table 1.* **Model accuracies on repeat prediction tasks.** The analyzed PLMs show high performance in different settings of repeat identification, motivating our research.

| Model | Synthetic | Identical | Approximate |
|---|---|---|---|
| **ESM-3** | 99.8% | 99.0% | 79.0% |
| **ESM-C** | 99.9% | 99.4% | 79.7% |

## 3  Locating a General Repeat Mechanism

To understand the mechanisms employed by PLMs to identify and complete each of the repeat types described in Section 2, we first identify the most important model components involved in repeat identification across all three tasks (Section 3.1). We then compare the resulting sets of components across tasks to assess whether the underlying mechanisms are shared or task-specific (Section 3.2).

### 3.1  Circuit Discovery and Evaluation

For each task variant, we locate a circuit $\mathbf{c}$: a minimal subset of interconnected model components that explain most of the model's performance on the task (Elhage et al., 2021). The circuit components are attention heads and MLPs.

To locate the components that matter the most for each task variant, we use attribution patching with integrated gradients (AP-IG; Hanna et al., 2024). For that, we work with pairs of sequences: an input sequence $s_i$ with a single masked token, and a counterfactual sequence $\hat{s}_i$. AP-IG

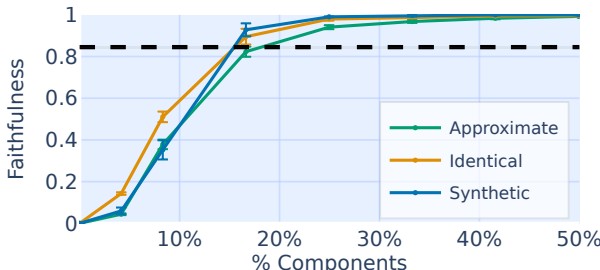

*Figure 2.* **ESM-3 circuit faithfulness scores.** Across the three tasks, the discovered circuits achieve high faithfulness (above the 85% threshold) using a small fraction of model components.

measures each component's importance by approximating how changing its activation from its value on $s_i$ to its value on $\hat{s}_i$ affects the log probability $L_{a_i}$ of the correct amino acid completion $a_i$ in the masked position. Intuitively, if a component is important then changing its activation should change the model's prediction toward the one it had on $\hat{s}_i$. For each task variant, we sample 500 sequences $s_i$ and construct each counterfactual $\hat{s}_i$ by replacing every amino acid in the non-masked repeat occurrence with its highest-matching BLOSUM62 substitution (Henikoff & Henikoff, 1992), a matrix that scores amino acid replacements based on evolutionary patterns. The example below shows an original (top) and counterfactual (bottom) sequence, with the masked position marked by an underscore:

```
...RLRHQLSFYG_PFALG...KLRHQLSFYGVAFALG...
...RLRHQLSFYG_PFALG...RIKYEIAYFAISYSIA...
```

We explore additional counterfactual types in Appendix G.

We rank components by absolute importance to construct increasingly large circuits and evaluate their faithfulness (Wang et al., 2023): the retained fraction of the full model's performance. For each circuit $\mathbf{c}$, we replace non-circuit component activations by their values for $\hat{s}_i$, and measure the log probability $L_{a_i}(s_i \to \hat{s}_i)$. We average over 500 evaluation examples, yielding $\overline{L}_a(s \to \hat{s})$, and normalize to compute the faithfulness of the circuit (Mueller et al., 2025):

$$F(\mathbf{c}) = \frac{\overline{L}_a(s \to \hat{s}) - \overline{L}_a(\hat{s})}{\overline{L}_a(s) - \overline{L}_a(\hat{s})}, \qquad (1)$$

where $\overline{L}_a(s)$ and $\overline{L}_a(\hat{s})$ are the average log probabilities of the correct token when none or all components are ablated, respectively. Thus, $F(\mathbf{c}) = 1$ indicates that the circuit fully preserves the original model performance on the repeat-completion task, while $F(\mathbf{c}) = 0$ indicates that the circuit performs equivalently to the counterfactual baseline. Appendix H contains further details on circuit discovery and evaluation.

**Results.** Fig. 2 shows the faithfulness results for the three task variants, averaged over five runs per task using differ-

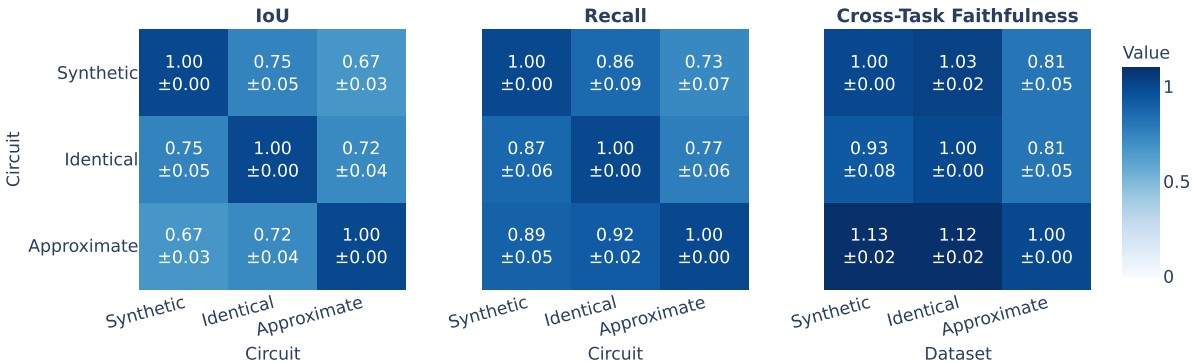

*Figure 3.* **Cross-Task Circuit Comparisons in ESM-3.** We compare the IoU, recall and cross-task faithfulness of the circuits found for the three repeat tasks. The IoU (left) shows relatively high overlap between all three tasks. The recall (middle) measures the fraction of the ground-truth circuit (x-axis) recovered by the predicted circuit (y-axis), and shows that the synthetic and identical circuits are largely subsumed within each other and within the approximate-repeat circuit. The cross-task faithfulness (right) denotes the faithfulness of the circuit (y-axis) on data from another task variant (x-axis), with scores normalized per target task relative to the faithfulness of its own circuit. The results show that the approximate repeat circuit is the only one that functionally generalizes to the other settings.

ent discovery and evaluation subsets. Following prior work (Hanna et al., 2025; Nikankin et al., 2025a), we consider a circuit as sufficient if its faithfulness is higher than 85%. Notably, the sufficient circuits for the synthetic and identical task variants on identical repeats are generally smaller (approx. 14%) than the sufficient circuits for the approximate repeats task (approx. 17%). This suggests that the model requires additional logic to succeed on the latter task.

### 3.2 Cross-Task Circuits Comparison

Having identified circuits for each task, we compare them across tasks to determine whether PLMs employ similar functionality for all three variants: synthetic repeats, natural identical repeats, and natural approximate repeats.

Following prior work (Hanna et al., 2025; Nikankin et al., 2025a), we measure structural overlap between circuits using intersection over union (IoU) of the components of each circuit. To account for size differences between circuits, we also measure the recall between their components to identify containment of one circuit within another. To complement these structural metrics, we measure functional equivalence using cross-task faithfulness (Hanna et al., 2025; Nikankin et al., 2025a)—the proportion of performance retained when a circuit $c_1$ is applied to the task used to find $c_2$ (e.g., using the circuit discovered for synthetic repeats to complete the approximate repeats task). The comparisons across pairs of tasks are repeated five times, based on circuits discovered using different random subsets of examples.

**Results.** As Fig. 3 shows, the cross-task comparison reveals high intersection between all three circuits (Fig. 3a), with particularly high recall for the approximate repeats circuit (Fig. 3b), indicating that it contains most components from the other two circuits. This structural finding is corrob-

orated by high cross-task faithfulness of the approximate repeats circuit (Fig. 3c). When evaluated on other tasks, the approximate repeats circuit achieves faithfulness over 1.0, indicating it functionally generalizes to sequences with identical repeats. In contrast, circuits discovered for the synthetic and identical repeats tasks do not fully recover performance on approximate repeats.

Taken together, these results suggest that a core mechanism is shared across all three tasks, with additional required functionality applied to complete sequences with approximate repeats. We thus focus the rest of our analysis on the circuit found for approximate repeats, as it appears to capture the broader mechanism employed across all tasks.

## 4 Individual Component Analysis

After identifying the circuit for the approximate repeats task, we aim to develop a deeper understanding of each component's role and how these components interact. In Sections 4.1 and 4.2, we characterize the functional roles of groups of attention heads and MLPs, respectively. In Section 5 we analyze how these component groups interact to identify and complete protein sequences with repeats.

### 4.1 Attention Heads

We restrict our analysis to attention heads that are consistently selected across multiple circuit discovery runs, focusing on components with systematic importance for task success, given the high variability in circuit discovery (Méloux et al., 2025). This yields 149 out of 1152 attention heads in ESM-3 and 102 out of 648 in ESM-C (see Appendix H.4).

To understand the functional roles of those individual heads, we first examine attention patterns across sample sequences. We identify three repeating attention behaviors (Fig. 4).

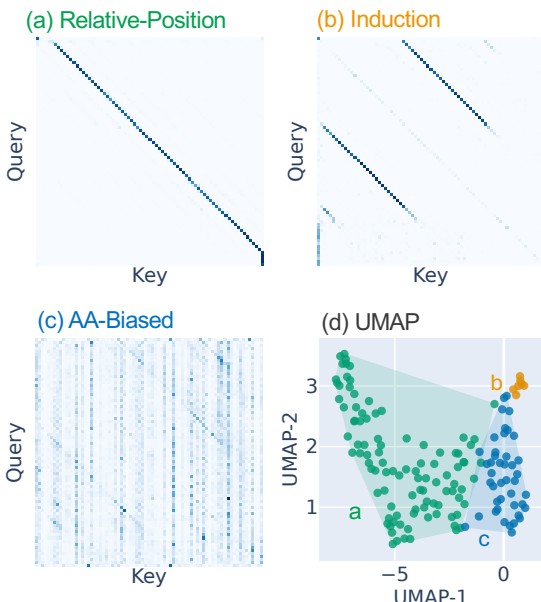

(a) Relative-Position  (b) Induction  (c) AA-Biased  (d) UMAP

*Figure 4.* **Active attention patterns in ESM-3.** (a–c) show example attention maps from three attention heads in the approximate-repeat circuit, for a single representative input: (a) fixed relative-position attention (diagonal); (b) induction attention between aligned repeat positions (two partial diagonals); and (c) amino-acid–biased attention (vertical). (d) Circuit attention heads are clustered and visualized using UMAP, colored by pattern type.

First, some heads attend to fixed positional offsets, either forward or backward along the sequence (Fig. 4a). Second, other heads exhibit induction-like patterns (Olsson et al., 2022), attending from tokens in one repeat to their aligned counterparts in the other repeat (Fig. 4b). Third, a subset of heads attend to specific groups of one or more amino acids, either globally across the entire sequence (Fig. 4c, vertical lines) or locally within a particular positional subset. See Appendix I.1 for additional examples of these patterns.

To rigorously quantify these patterns and identify additional potential head behaviors, we define several features for each attention head and cluster them based on these features. These features capture key patterns in attention maps and outputs, and are summarized in Table 2, with full definitions and motivations provided in Appendix I.2.

We compute these features across the approximate repeats dataset and cluster the heads using k-means (Lloyd, 1982). We find $k = 3$ clusters to be optimal via elbow and silhouette analyses for ESM-3 (Appendix I.3), where each cluster (visualized in Fig. 4) aligns with one of the concepts from the manual analysis: a majority cluster of 99 heads with high attention to fixed positional offsets, a medium-sized cluster of 43 heads biased toward subsets of amino acids,[3] and a small cluster of induction heads. Relative-

---

[3]A subset of amino-acid–biased heads also exhibit a high repeat-

*Table 2.* Attention head features. Full details in Appendix I.2.

| Feature | Description |
| --- | --- |
| Induction score | Pattern-matching and copying across aligned repeats. |
| Relative position | Consistent attention at fixed offsets. |
| Amino acid bias | Selectivity for specific amino acids. |
| Repeat focus score | Sharper attention to repeat regions. |
| Attention sharpness | Attention distribution concentration. |
| Boundary attention | Attention to boundary tokens. |
| Head contribution | Contribution to the final prediction. |

position heads span all layers, while induction heads and amino-acid–biased (AA-biased) heads are concentrated in middle and later layers. A full cluster analysis is provided in Appendix I.4.

While induction heads and relative-position heads were previously linked to repeat completion in language models (Elhage et al., 2021), the AA-biased heads reflect sensitivity to the protein data on which PLMs are trained, indicating a contribution of biological information to task success.

### 4.2 MLP Neurons

Unlike attention heads, where the discovered circuits only included a small fraction of them, the approximate-repeat circuit includes most MLP blocks across all layers.[4] To understand their functional roles, we analyze these MLPs at the neuron level. We define a neuron as a single scalar dimension in the post-activation vector (Geva et al., 2021), with the formal definition provided in Appendix J.1.

We identify the most important neurons for the task by applying AP-IG at the neuron level. We compute an importance score for each neuron in each MLP using the same procedure from Section 3.1. We then measure the faithfulness of the circuit when ablating non-important neurons: We start from the discovered circuit—which includes the full MLP blocks—and progressively prune neurons from each layer. For a given percentage $p$, we retain the top-$p$% neurons (ranked by absolute AP-IG score) in each circuit MLP layer, as well as all circuit attention heads. Neurons outside this subset, as well as all components outside the original circuit, are ablated, and faithfulness is calculated as in Equation (1).

The results (Fig. 5) show that retaining as little as roughly 3% of the neurons per MLP layer in the original circuit is sufficient for recovering almost all circuit performance. In line with the attention head analysis, we focus on interpreting neurons that are consistently selected as important across random seeds. We end up with 3,394 and 3,510 neurons for ESM-3 and ESM-C, respectively (1.7% and 3.17% of all neurons; see Appendix H.4 for additional details).

---

focus score; further details are provided in Appendix I.5.

[4]This is not uncommon also in text LMs (Wang et al., 2023).

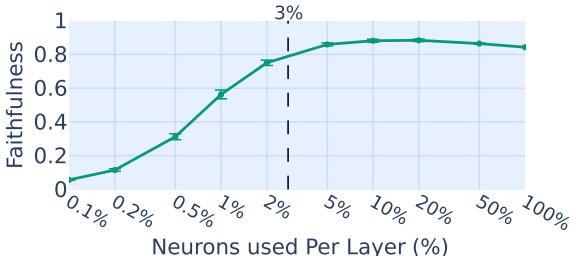

*Figure 5.* **Circuit faithfulness of subsets of neurons in ESM-3.** 3% of MLP neurons per layer are sufficient to recover most circuit faithfulness (80/85%), indicating sparsity in the neuron basis.

**Qualitative examples.** As an initial exploratory step, we visualize the activation patterns of high-scoring neurons. Observing these activations reveals several recurring patterns, exemplified in Fig. 6a: First, some neurons exhibit a bias toward specific amino acids. Second, other neurons activate on subsets of amino acids that share similar biochemical properties or exhibit high evolutionary substitution likelihood according to BLOSUM62.[5] Third, some neurons activate more strongly within repeated segments. These behaviors vary in their consistency: some amino-acid–selective, subset-selective, or repeat-focused neurons activate consistently across all relevant token occurrences, whereas others activate only in specific sequence contexts. Additional examples of neuron activation patterns are provided in Appendix J.4.

**Quantitative analysis.** To systematically characterize neuron behavior beyond manual inspection, we define a set of concepts that capture these observed patterns. Each concept represents a binary classification over tokens—a token belongs to a concept if it satisfies the defining property (e.g., the concept *positively charged amino acids* includes tokens corresponding to amino acids R, K, or H).

The concept set comprises three main categories: a concept for tokens within repeated regions; a concept for each of the 20 standard amino acids; and concepts representing subsets of amino acids with shared biochemical properties. For the latter, we group similar amino acids based on properties such as hydrophobicity, volume, and charge using established biochemical groupings from the IMGT physicochemical classification scheme (Pommie et al., 2004). We also consider groupings of amino acids based on secondary-structure tendencies (Chou & Fasman, 1978), as well as groupings derived from the BLOSUM62 substitution matrix by identifying amino-acid groups that are more likely to substitute for one another. The full list of concepts and their definitions appears in Appendix J.2.

For each pair of neuron and concept, we partition tokens into those belonging to the concept and those that do not. Then, we measure how well the neuron's activation values discriminate between these two groups, by scanning possible decision thresholds and calculating the area under the receiver operating characteristic curve (AUROC). Each neuron is assigned to the concept with which it achieves the highest AUROC score, following similar approaches for matching concepts to SAE features in PLMs (Nainani et al., 2025; Simon & Zou, 2025). Additional details on the neuron–concept scoring procedure, including baseline construction, are provided in Appendix J.3.

Fig. 6b shows AUROC score distributions for each neuron's best-matching concept. Overall, early-layer neurons achieve higher AUROC under our concept set than mid–late layers, while almost all neurons still perform far above random. Across all layers, neurons are most frequently matched with singleton amino-acid concepts, followed by BLOSUM62 groups and other concepts based on biochemical similarity. In mid-to-late layers, a substantial fraction of neurons are matched with the repeat-region concept. Notably, these repeat-matched neurons emerge in approximately the same layer range as the induction heads, suggesting a connection between induction mechanisms and repeat-related neuron specialization—a phenomenon also observed in text LMs (Hiraoka & Inui, 2025; Doan et al., 2025).

ESM-C exhibits similar overall patterns (Appendix F.4). However, we observe several differences between the models (Fig. 6c) when considering neurons matched with concepts derived from IMGT classifications and secondary-structure propensities. In particular, ESM-3 neurons are matched with more such concepts, including multiple neurons corresponding to helix-breaking amino acids like glycine (G) and proline (P).[6] In contrast, no such neurons are observed in ESM-C. This difference may be related to ESM-3's additional training on structural and functional information, which allow it to capture further biochemical properties and relationships between amino acids.

Overall, these results reveal two complementary groups of task-relevant neurons: biologically oriented neurons that encode amino acid identity and biochemical similarity, and repeat-related neurons likely involved in induction mechanisms. While the presence of neurons capturing amino-acid similarity is expected, it is notable that many important neurons for the approximate repeat task are associated with BLOSUM62 groups, which are central to classical alignment scoring. This suggests that the internal similarity function learned by PLMs aligns with substitution-based

---

[5]In Fig. 6a, we observe a neuron that activates strongly on I and V, two amino acids that frequently substitute one another and share similar biochemical properties (both are uncharged, hydrophobic, and aliphatic) (Nelson & Cox, 2005; Pommie et al., 2004).

[6]An $\alpha$-helix is a common spiral-shaped structural motif in proteins. Glycine and proline are known as "helix breakers" because their geometric constraints disrupt the regular helical structure, making helices unstable when these amino acids are present.

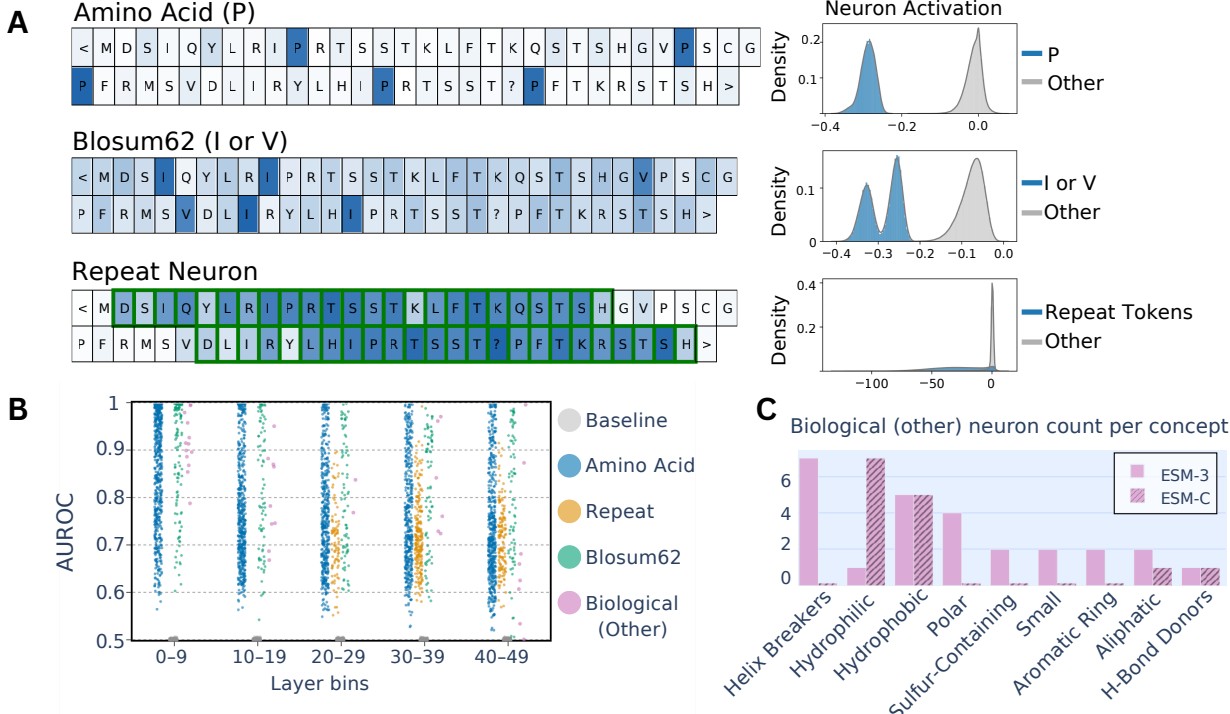

*Figure 6.* **(A)** Examples of three task-relevant neurons in ESM-3: an amino-acid–specific neuron, a neuron that responds to substitutable amino acids according to BLOSUM62 groupings, and a repeat-focused neuron. We show token-level activations on an example sequence (left) and activation distributions over the full dataset (right). **(B)** AUROC scores for each neuron's best-matching concept across ESM-3 layers. Each point is a neuron, plotted by layer and AUROC, and colored by concept category. "Biological (Other)" denotes groupings from IMGT physicochemical classes and secondary-structure propensities. **(C)** Number of neurons associated with "Biological (Other)" concepts in ESM-3 and ESM-C, restricted to neurons with AUROC $\geq 75\%$.

similarity used in traditional repeat alignment methods.

# 5 The Repeat Detection Mechanism

The previous section described the main component groups involved in PLM repeat detection: induction heads, relative-position heads, AA-biased heads, and various MLP neuron groups. We now explain how these components interact and promote the correct prediction. We first measure mutual influence between component groups, then assess each group's contribution to the final prediction at each layer.

To quantify interactions between component groups, we apply EAP-IG (Hanna et al., 2024). For each pair of component groups, we average the interaction effects over all corresponding component pairs, focusing on positive interactions (see Section H.5 for further details).

The results (Fig. 7) show that induction heads and MLPs exert the strongest direct influence on model output. AA-biased heads have a smaller effect, and relative-position heads show the weakest direct contribution. The input strongly affects relative position-heads, AA-biased heads, and MLPs, but has minimal direct influence on induction heads, suggesting they are influenced mainly after intermediate input processing. Additionally, strong interactions

| Src components | AA-biased heads | Rel. Pos. heads | Induction heads | MLPs | Output logits |
|---|---|---|---|---|---|
| Input Embeddings | 3.91 | 24.72 | 0.82 | 11.75 | 0.00 |
| AA-biased heads | 0.25 | 0.16 | 0.85 | 1.06 | 8.52 |
| Rel. Pos. heads | 0.18 | 0.24 | 1.56 | 4.18 | 2.63 |
| Induction heads | 0.47 | 1.00 | 1.92 | 5.38 | 40.94 |
| MLPs | 0.69 | 0.52 | 2.02 | 1.55 | 65.51 |

Dst components

*Figure 7.* **Aggregated interactions between component groups in ESM-3.** Larger scores indicate stronger influence. Induction heads and MLPs show the strongest influence on the logits, with weaker contributions from AA-biased heads. The input primarily affects non-induction components, and interactions are strongest between attention heads and MLPs, highlighting the central role of MLPs in mediating information flow.

between attention heads and MLPs indicates that much inter-component communication flows through the MLPs.

Next, we analyze when and by which component groups the correct token is promoted at the masked-token position using the logit lens (nostalgebraist, 2020), which projects intermediate representations into the vocabulary space. We

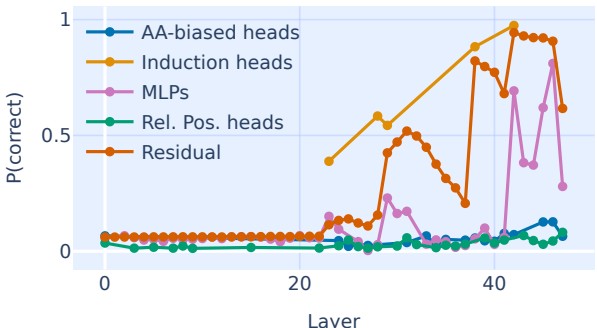

*Figure 8.* **Logit-lens analysis of ESM-3 component outputs.** We measure the direct averaged contribution of each component group to the correct prediction at each layer. Induction heads and late MLPs are the major promoters of the correct answer, while other component groups only contribute indirectly.

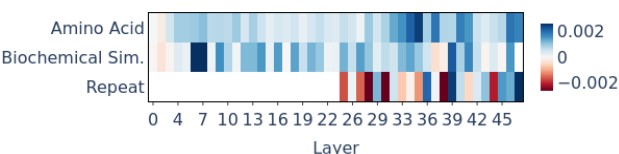

*Figure 9.* **Attribution analysis of neuron-group importance across layers in ESM-3.** Biochemical-similarity neurons are most important in early layers, amino-acid neurons contribute stronger in later layers, and repeat-related neurons often show inhibitory effects, reflected by negative contributions across layers.

apply the logit lens to the masked token's representation at the end of each layer (residual stream), as well as to the outputs of MLPs and attention heads. For attention, we aggregate heads according to the three previously identified groups and report each group's average at every layer..

As shown in Fig. 8, induction heads are the first component group to promote the correct token in the middle layers. Other attention heads do not contribute directly at these layers but act as mediators, aligning with our previous findings. In contrast, MLPs contribute to the correct prediction in later layers, downstream of induction heads. The final MLP layer seems to reduce confidence in the correct output token, a role also observed in LLMs (Lad et al., 2024).

Finally, we plot the average importance scores (from Section 4.2) for all neurons linked to concepts with an AUROC above 0.75, averaged over neuron groups. We refer to neurons encoding features related to IMGT, secondary structure, and BLOSUM62 as biochemical similarity neuron group, along with single amino-acid neuron group and repeat neuron group. The results (Fig. 9) show that amino-acid neurons are more important in later layers, while biochemical similarity neurons are more important in early layers, although both groups show high effects across most layers. Repeat neurons only impact mid-to-late layers, often serving an inhibitory role shown by their negative effects.

**Summary of the repeat-detection mechanism.** Taken together, these analyses suggest a plausible computational path for repeat detection. First, relative-position heads and MLPs–primarily biochemical similarity neurons and amino acid neurons–build contextual representations that align tokens across the two repeats (Fig. 1, Stage I). Second, induction heads exploit these representations by attending from the masked position to its aligned token and copy the corresponding residue, thereby increasing the probability of the correct token. In these layers, repeat-related neurons play an inhibitory role (Fig. 1, Stage II). Finally, late MLPs driven mainly by amino acid and repetition neurons shape the output distribution, while AA-biased heads have a weaker direct contribution to the logits (Fig. 1, Stage III).

## 6 Related Work

**Interpretability of PLMs.** Prior work has explored pre-hoc approaches that build intrinsically interpretable models (Ismail et al., 2025) and post-hoc methods that analyze trained models; we take a post-hoc approach. Early work found that attention patterns capture structural and functional protein properties (Rao et al., 2021; Vig et al., 2021), while recent studies used sparse auto-encoders to find interpretable PLM features (Simon & Zou, 2025; Adams et al., 2025; Parsan et al., 2025; Gujral et al., 2025; Liu et al., 2026). In contrast, we show that individual neurons are often interpretable without decomposition. Nainani et al. (2025) traced circuits for residue contact prediction in two single proteins; we apply circuit analysis to understand how PLMs identify and complete repetitions across diverse protein sequences. In concurrent work, Zhang et al. (2026) study pathological repetition in PLM generation and propose a steering-based mitigation method. Our findings provide a mechanistic perspective on repeat-related components that may be involved in this phenomenon.

**Repeat identification in LLMs.** Several mechanisms enable LLMs to identify and complete repeating sequences. Induction heads—attention heads that detect repeated patterns and copy their continuation (Olsson et al., 2022)—drive this behavior in exact and semantic copying (Feucht et al., 2025) and in in-context learning (Crosbie & Shutova, 2025). Repetition neurons, which activate strongly during repeat sequence generation (Hiraoka & Inui, 2025) also contribute to such completions. We show such mechanisms generalize beyond LLMs to identify repeats in PLMs, where they combine with biologically specialized components that identify and promote biological features.

## 7 Conclusions and Limitations

How do PLMs identify repeating sequences, even when these sequences are not identical? Using mechanistic analysis and circuit discovery, we find that repeat detection is driven primarily by induction heads and other mechanisms

familiar from LLMs, combined with biologically specialized components such as neurons selective for biochemically similar amino-acid groups. We also uncover similarities and differences between two prominent PLMs.

Our analysis is a first step toward understanding this key mechanism in PLMs, but has several limitations. We focus on sequence-level repeats with exactly two occurrences and up to 50% substitution rate, while natural proteins can exhibit more repeat occurrences or repeat patterns in structure rather than sequence. In addition, our component analysis carries inductive bias due to manually defined concepts and clustering metrics, potentially missing other relevant behavioral patterns, which could be addressed by automated discovery methods.

## Impact Statement

This paper presents work whose goal is to advance the interpretability of protein language models. We do not think any potential societal consequences should be specifically highlighted here.

## Acknowledgments

We are grateful to Stefan Huber and Nadav Brandes for providing feedback. YN was supported by the Council for Higher Education (VATAT) Scholarship for PhD students in data science and artificial intelligence. AR was supported by the Azrieli international postdoctoral fellowship and the Ali Kaufman postdoctoral fellowship. This research was supported by Coefficient Giving, the Israel Science Foundation (grant No. 2942/25), and the European Union (ERC, Control-LM, 101165402). Views and opinions expressed are however those of the author(s) only and do not necessarily reflect those of the European Union or the European Research Council Executive Agency. Neither the European Union nor the granting authority can be held responsible for them.

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

# A   Protein Repeats: Evolutionary Background

Sequence repeats play a significant role in protein evolution. As a result of unequal recombination, a DNA-copying error that occurs when two similar chromosomes align incorrectly and swap unequal pieces, genomic regions can be duplicated. This may result in duplication of a full protein or in insertion of a sequence repeat within a protein sequence. Since the original sequence and its functional role are maintained, the new repeat is free to evolve into a new functionality under reduced evolutionary constraints, while dosage maintenance (the need to keep the amount of a gene or protein produced at a functional, non-toxic level) and folding constraints might actually accelerate evolution away from the original sequence and its functionality. Upon generation of the first repeat, the addition of more repeats is facilitated by crossover at identical sequence repeats (Andrade et al., 2001).

Repeats are often used as scaffolds, and certain proteins use domain repeat number variation (changes in how many times a specific functional domain is repeated within a protein) to adapt to changing environments (Verstrepen et al., 2005). In particular, prokaryotic proteins continuously use domain repeat variation, as indicated by the significantly higher sequence identity between repeats, in contrast to eukaryotic proteins (Reshef et al., 2010). In contrast to slow evolution by incremental point mutations, repeat insertion thus provides the basis for fast evolutionary adaptation, but uncontrolled propagation of sequence repeats often also leads to disease (Delucchi et al., 2020).

Thus, repeats in proteins represent a key evolutionary mechanism, motivating our study of how protein language models detect such patterns.

# B   Dataset Curation

We construct three datasets containing protein sequences with repeating segments: a synthetic dataset composed of randomly generated amino acid sequences and two datasets of naturally-occuring proteins—one of sequences with identical repeat segments and the other of sequences with approximate repeat segments.

**Synthetic repeats.**   We generate a dataset of 20,000 synthetic protein sequences, each of length 200 amino acids, using the following template:

$$[\text{Repeat segment}] \, | \, [\text{Spacer}] \, | \, [\text{Repeat segment}] \, | \, [\text{Random remainder}].$$

For each sequence, we sample a segment of length $l_1 \in [10, 30]$ comprised of random amino acids, and insert it twice as the repeated segment. The two occurrences are separated by a spacer of length $l_2 \in [0, 50]$ of random amino acids. The remainder of the sequence is filled with randomly sampled amino acids to reach a total length of 200. We generate an equal number of sequences for each combination of repeat length and spacer length.

**Natural repeats.**   We sample 10 million protein sequences from UniRef50 (Suzek et al., 2014), restricting sequence lengths to 50–400 amino acids to ensure feasibility of circuit discovery. We identify repeats within proteins using RADAR (Heger & Holm, 2000), an algorithm that detects *de novo* repeats (i.e., without relying on known repeat templates) by identifying internal sequence symmetries through self-alignment of the protein sequence. To do so, we first mask low-complexity regions with SEG (Wootton, 1994) in each protein, which are segments with low amino-acid diversity, often dominated by repetitions of a few amino acids or short-period repeats that can confound the detection of biologically meaningful duplications. We run the RADAR algorithm on the masked proteins to identify repeat regions. We then partition sequences with detected repeats into identical and approximate repeats and apply the following filters:

- **Overlapping repeats.** We remove sequences in which the identified repeats are overlapping (i.e., share at least one residue).

- **More than two repeat segments.** We remove sequences with more than two repeating segments.

- **Extreme repeat length** We remove sequences with extremely short (below the 5th percentile) or extremely long (above the 95th percentile) repeats.

- **High-mutation approximate repeats** For approximate repeats, we remove sequences whose repeats have more than 50% mutations relative to one another.

- **Identical sub-repeats.** RADAR reports only statistically significant repeated segments. However, the sequence may contain segments identical to a sub-string of an identified repeat, that can influence the model's prediction. Thus, we apply an additional identical-repeat detection procedure (Banerjee et al., 2008) and remove any sequences that contain an identical sub-string of the RADAR-identified repeat outside the boundaries of the two repeat occurrences.

In case the algorithm identifies several distinct repeats with a single sequence, we keep the sequence as long as one of the distinct repeats passes the filters. To balance the amount of approximate repeats across similarity levels $s$ (the percentage of identical amino acids), we retain all approximate repeats with $s \geq 75\%$ and sample an equal number from repeats with $s < 75\%$. The filtering process yields $5,317$ sequences with identical repeats and $17,939$ sequences with approximate repeats.

## C    Additional Repeat Type and Examples

### C.1    Insertions/Deletions Task Definition

In addition to the approximate-repeat task defined in the main paper for substitution-only repeats, we also consider approximate repeats that differ due to insertion and deletion (indel) mutations. An indel mutation corresponds to the insertion of one or more amino acids into a sequence or the deletion of existing amino acids, resulting in a gap when aligning the two repeat segments, i.e., amino acids in one segment have no matching counterpart in the other. We use approximate-repeat pairs obtained during dataset curation that contain both substitutions and indels. We mask a position that is identical to its aligned counterpart and is adjacent to a gap in the alignment. The gap disrupts local sequence continuity, making direct pattern matching more difficult. When measuring model performance on sequences with indel mutations, both models achieved incredibly low accuracy (ESM-3 and ESM-C achieve $42.9\%$ and $48.1\%$ accuracy on these sequences, respectively). Thus, we focused on sequences without indel mutations in our circuit analysis.

### C.2    Masked Input Examples

Table 3 shows examples of repeats and their alignments for all types of analyzed repeats.

*Table 3.* **Examples of masked inputs for each task.** *Masked Input* shows the masked residue as an underscore; ellipses (`...`) denote non-repeat regions not shown for brevity. *Repeat Alignment* marks exact matches (`|`), substitutions (`.`), and gaps (`−`). *Eligible Positions* specifies which repeat positions may be masked under each task definition.

| Task | Masked Input | Repeat Alignment | Eligible mask positions |
|---|---|---|---|
| Synthetic | `...NFDIQREHYKTEWYN...` `...NFDI_REHYKTEWYN...` | `NFDIQREHYKTEWHRYN` `|||||||||||||||||` `NFDIQREHYKTEWHRYN` | All positions |
| Identical | `...MKTAYIAKQRQISFVHFS...` `...MKTA_IAKQRQISFVHFS...` | `MKTAYIAKQRQISFVKSHFS` `||||||||||||||||||||` `MKTAYIAKQRQISFVKSHFS` | All positions |
| Approximate (Substitution-Adjacent) | `...PYTSAVTQTR...` `...PYRA_VTQTR...` | `PYTSAVTQTR` `||..||||||` `PYRAAVTQTR` | Identical positions adjacent to substitutions (Y, A in example) |
| Approximate (Indel-Adjacent) | `...GNAASATKLQTSR...` `...GNAASAT_ATQDSR...` | `GNAASATK−LQTSR` `|||||||| .|.||` `GNAASATKATQDSR` | Identical positions adjacent to gaps (K in example) |

## D    Task Accuracy Evaluation

In each setting, we evaluate model accuracy by masking each task-eligible position per repeat, as defined in Table 3. We mask and evaluate each eligible position independently. For every masked position, we record whether the model correctly predicts the masked amino acid.

We define the average accuracy as the fraction of eligible positions within a repeat for which the model predicts the correct amino acid. We first compute this metric per repeat, then average it across all repeat entries belonging to the same task.

Note that while the task definitions for circuit analysis involve sampling a single masked position per repeat, we conduct performance evaluation over all eligible positions. This evaluation helps identify repeats that the model reliably identifies as repeats prior to mechanistic analysis.

# E   Failure Analysis for the Approximate-Repeat Task

In Section 2, we show that PLMs achieve approximately 79% accuracy on the approximate-repeat task. To better understand these failures, we analyze performance along two factors known to make sequence-based repeat detection more challenging: shorter repeat length and greater divergence between repeat occurrences (Andrade et al., 2001; Jorda & Kajava, 2009).

We group approximate-repeat examples into bins according to repeat length and substitution rate. Repeat length is measured as the number of residues in each repeat occurrence, and substitution rate is measured as the fraction of aligned repeat positions that differ between the two occurrences. We then compute the average accuracy within each bin. As shown in Table 4, failures are concentrated in short repeats with high substitution rates, while performance improves substantially as repeat length increases. This pattern is consistent across both ESM-3 and ESM-C, suggesting that short and highly divergent repeats constitute a shared failure mode across PLMs.

*Table 4.* **Approximate-repeat accuracy by repeat length and substitution rate.** Accuracy is averaged within each bin.

| Model | Subst. rate | Repeat length | | | | |
|---|---|---|---|---|---|---|
| | | **1–10** | **11–15** | **16–20** | **21–25** | **26+** |
| ESM-3 | 0–25% | 0.77 | 0.94 | 0.96 | 0.98 | 0.98 |
| | 25–50% | 0.45 | 0.63 | 0.85 | 0.92 | 0.96 |
| ESM-C | 0–25% | 0.75 | 0.94 | 0.96 | 0.96 | 0.97 |
| | 25–50% | 0.51 | 0.65 | 0.85 | 0.92 | 0.94 |

The bin-level analysis above shows that the model tends to fail when predicting masked tokens in short and highly divergent repeats. However, it does not directly tell us whether this degradation reflects a weakening of the repeat-completion mechanism itself, identified in Section 5.

To test this, we focus on induction heads, which we characterized as components that align repeat positions and are the first components to promote the correct answer at the masked token, thereby contributing directly to task success. We use natural proteins containing identical repeats of lengths 10, 20, and 30, randomly sampling 40 examples for each repeat length. For each sample, the model is tasked with predicting a masked token within one repeat occurrence, as in the original identical-repeat setting. We progressively mutate the other occurrence using the most likely BLOSUM-based substitutions, while keeping the amino acid aligned with the masked position unchanged. At each substitution rate, we measure the average AP-IG attribution assigned to induction heads. Since these heads are key components for repeat-based masked-token prediction, reduced attribution suggests that the mutated occurrence becomes less useful to the model as a reliable repeat-based signal.

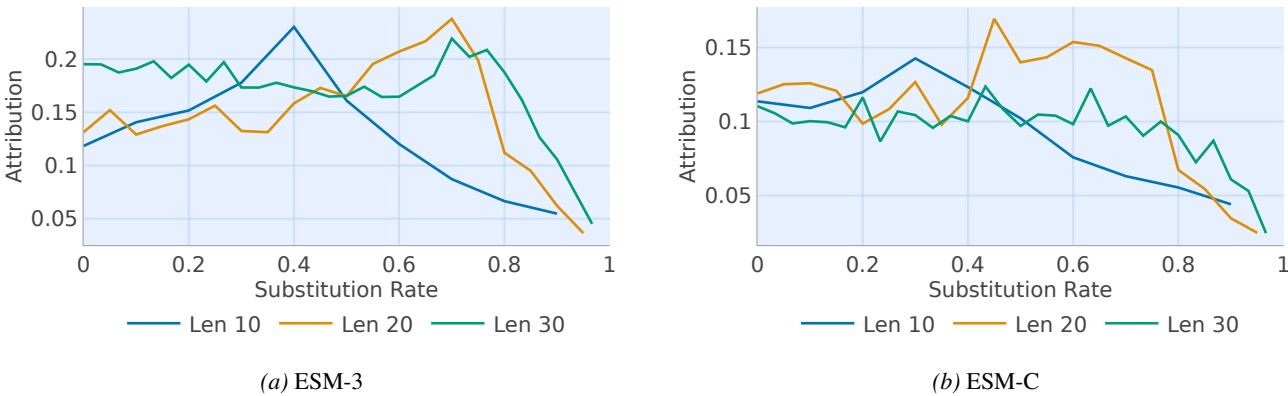

*(a)* ESM-3             *(b)* ESM-C

*Figure 10.* **Effect of substitution rate on induction-head contributions to repeat completion.** In both models, induction-head attribution remains stable or initially increases up to a length-dependent substitution rate, after which it decreases sharply. This transition occurs earlier for shorter repeats, suggesting that the induction-based repeat-completion mechanism is less robust when the repeat signal is short or highly divergent.

**Results.** Fig. 10 shows a threshold-like pattern: induction-head attribution remains stable or initially increases up to a length-dependent substitution rate, after which it drops sharply. This transition occurs earlier for shorter repeats: around a $30-40\%$ substitution rate for length 10, compared to approximately $70\%$ for length 20 and $75\text{–}80\%$ for length 30. These results suggest that the induction-head mechanism can support repeat completion under substantial divergence when repeats are sufficiently long, but becomes less effective when the repeat signal is short or highly divergent. This is consistent with the approximate-repeat failure analysis and provides a possible mechanism-level explanation for the weaker performance on short and divergent repeats.

## F  ESM-C Results

We report the results for all experiments performed on ESM-C (ESM Team, 2024).

### F.1  Circuits Faithfulness Results

We repeat the circuit discovery and evaluation process (Section 3.1) for ESM-C. Fig. 11 shows that across repeat settings, approximately $25\%$ of ESM-C model components are required to reach the $85\%$ faithfulness threshold. This differs from the results reported for ESM-3, where the approximate repeats circuit contained a larger fraction of components than the synthetic and identical repeats circuits.

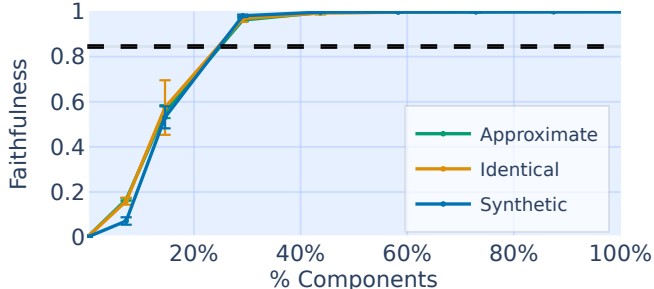

*Figure 11.* **ESM-C circuit faithfulness scores.** Across all three task settings, the discovered circuits achieve high faithfulness using approximately $25\%$ of model components.

### F.2  Cross-Task Circuits Comparison

We repeat the cross-task circuit comparisons (Section 3.1) for ESM-C. Fig. 12 shows the results mirror the generalization patterns we observe for ESM-3. The approximate repeats circuit is the only circuit that functionally generalizes to both

synthetic and identical repeat sequences in terms of cross-task faithfulness, while all three task settings show similar structural overlap.

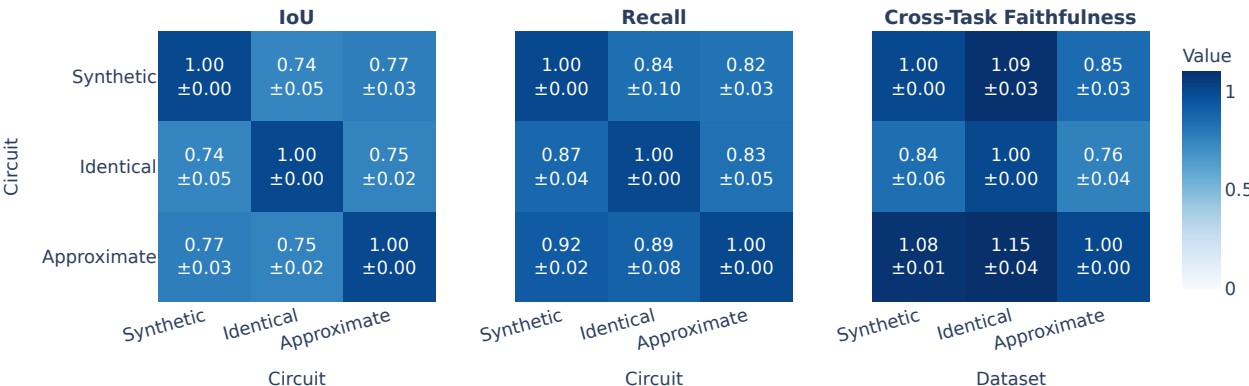

*Figure 12.* **Cross-task circuit comparisons in ESM-C.** All circuits show similar structural overlap, while only the approximate repeats circuit exhibits strong functional generalization to the other tasks.

## F.3 Attention Heads Clustering

Manual inspection of attention patterns for heads in the approximate repeats circuit reveals the same recurring patterns we observe in ESM-3, as shown in Fig. 13. See Appendix I.1 for additional examples.

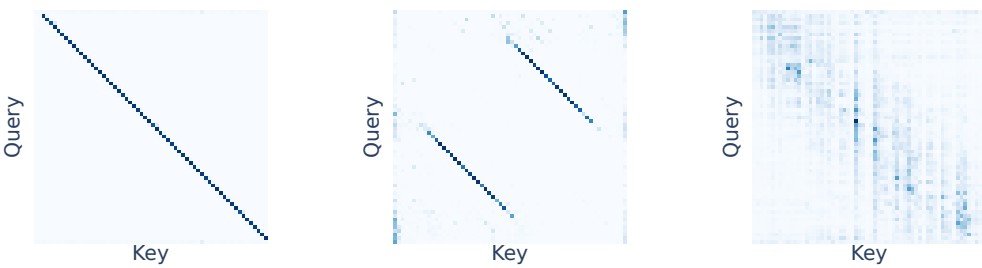

*Figure 13.* **Attention patterns in ESM-C.** Example attention maps from three attention heads in ESM-C approximate repeats circuit. for a single representative input, illustrating the same recurring patterns we observe in ESM-3: **Left:** fixed relative position attention, producing a diagonal pattern; **Middle:** induction attention between aligned repeat positions, resulting in two partial diagonals due to bidirectional alignment; **Right:** amino acid biased attention, producing a vertical pattern.

We apply the same clustering procedure from Section 4.1, using the same attention-based features as for ESM-3. For ESM-C, the Elbow method indicates an optimal choice of $k = 3$, whereas Silhouette analysis favors $k = 2$. We select $k = 3$ as it separates relative-position heads from amino-acid–biased heads, yielding more interpretable clusters (Appendix I.3).

With $k = 3$, each cluster aligns with one concept from the manual inspection: a majority cluster of 62 heads attending to fixed positional offsets, a medium-sized cluster of 28 heads biased toward subsets of amino acids, and a small cluster of 12 induction heads. Relative-position heads are distributed across all layers, whereas induction heads and amino-acid–biased heads are concentrated in the middle and later layers. Full clustering results are reported in Appendix I.4. Fig. 14 shows a UMAP visualization of the clusters.

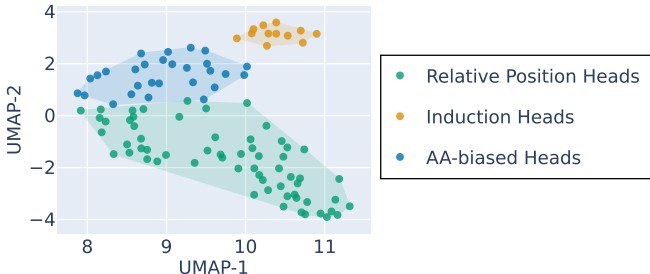

*Figure 14.* UMAP projection of clustered attention heads in ESM-C, colored by pattern type.

## F.4 Neuron Classification

We find MLP neurons important to the approximate repeat task in ESM-C and rank them (as done in Section 4.2). Results (Fig. 15) show that 7% of the MLP neurons per layer recover most of the circuit faithfulness, showing a slight increase from the required percentage in ESM-3.

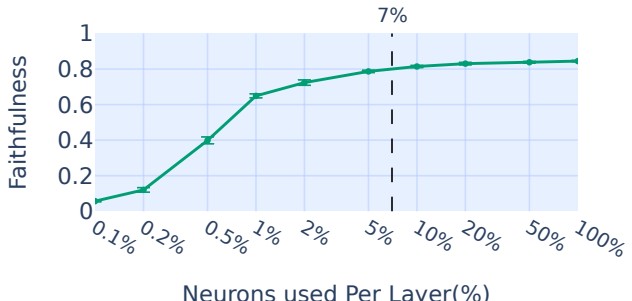

*Figure 15.* **Circuit faithfulness with a subset of neurons in ESM-C.** Retaining approximately 7% of MLP neurons per layer achieves 80% faithfulness.

Manual inspection of important neuron activations in ESM-C (see Appendix J.4 for examples) reveals patterns similar to those observed in ESM-3: (i) neurons that exhibit a bias toward specific amino acids; (ii) neurons that activate on subsets of amino acids with similar biochemical properties or with high substitution likelihood according to BLOSUM62; and (iii) neurons that activate more strongly within repeated segments. Thus, we apply the classification procedure from Section 4.2 to ESM-C neurons.

The results (Fig. 16) are similar to ESM-3. Across all layers, neurons are most frequently matched to singleton amino acid concepts, followed by BLOSUM62 concepts. In mid-to-late layers, a substantial fraction of neurons are matched to the repeat-region concept. Similar to ESM-3, early-layer neurons achieve higher AUROC than mid-to-late layers, and most neurons perform far above random.

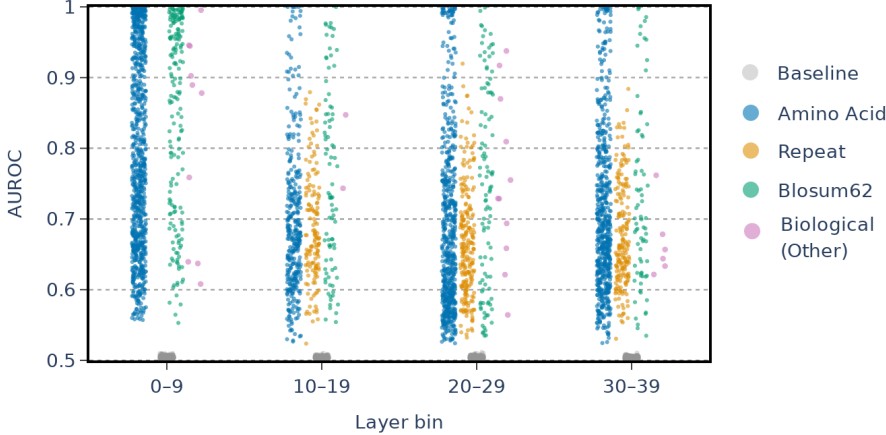

*Figure 16.* **AUROC scores for each neuron's best-matching concept across ESM-C layers.** Each point corresponds to a neuron, plotted by layer and AUROC score, and colored by concept category. "Biological (Other)" denotes groupings from IMGT physicochemical classes and secondary-structure propensities.

## F.5 Interaction Between Groups of Components

We measure interactions between component groups in the ESM-C approximate repeats circuit (as described in Section 5 and Section H.5).

Results (Fig. 17) show that, consistent with ESM-3, induction heads and MLPs exhibit the strongest direct influence on the output logits. We also observe a strong influence of the input on relative-position heads and MLPs. Interactions between induction heads and MLPs are relatively high, supporting a tight coupling between these components, consistent with patterns observed in ESM-3. In contrast to ESM-C, the input exerts a direct, relatively high influence on induction heads.

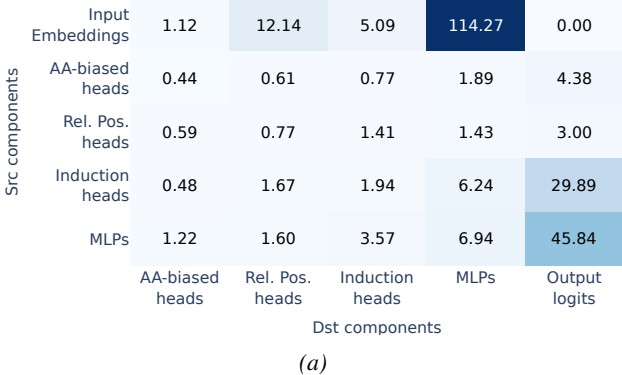

*(a)*

*Figure 17.* **Aggregated interactions between component groups in ESM-C.** Positive interactions between source and destination component groups, reflecting connections that support task performance.

## F.6 Direct Contributions to Prediction

We measure the direct contributions of each ESM-C component group to the prediction of the correct answer (as described in Section 5). The results (Fig. 18) show some consistency with ESM-3. Namely, induction heads are the first component group to promote the correct token, emerging in the middle layers. These are followed by the contribution of MLPs to the correct prediction. A single large difference emerges between the models: in ESM-3, final-layer MLPs also show inhibitory activations, modulating the prediction confidence. In contrast, ESM-C MLPs only increase the confidence in the predicted answer in the final layers.

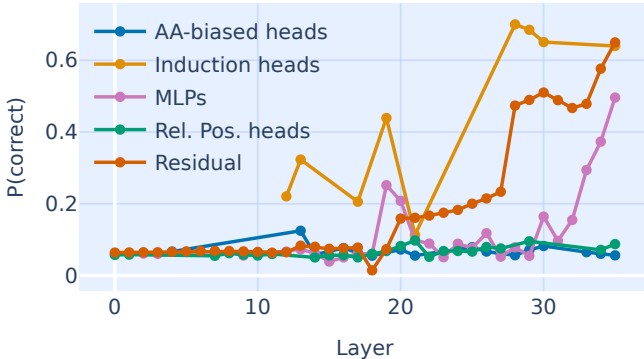

*Figure 18.* **Logit-lens analysis of ESM-C component outputs.** Induction heads promote the correct token in the middle layers, while final-layer MLPs continue to increase the prediction confidence.

Following our analysis in Section 5, we plot the average importance scores for all circuit neurons with AUROC scores above 0.75, aggregated by neuron group. The results (Fig. 19) show that, consistently with ESM-3, many repeat-related neurons exhibit inhibitory effects. Another similarity is the strong positive attribution shown by amino-acid and biochemical neurons in early-to-mid layers.

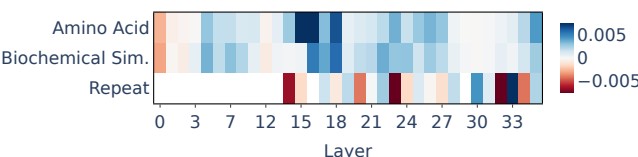

*Figure 19.* **Attribution-based analysis of neuron-group importance across layers in ESM-C.** Biochemical similarity and amino-acid neurons are most important in mid layers, with amino-acid neurons contributing more strongly in the final layer. Repeat-related neurons exhibit inhibitory effects, indicated by negative contributions across multiple layers.

## G    Counterfactual Selection

In section Section 3.1 we describe the usage of counterfactual sequences in circuit discovery, and present results for counterfactuals built by substitution of each amino acid with a similar amino acid, based on the BLOSUM62 matrix. In this section we describe several alternative strategies to counterfactual construction and the resulting circuits for each alternative.

### G.1    Counterfactual Design

To identify components responsible for repeat identification and prediction, we define counterfactuals that approximate how the model would behave if no repeat were present. Our guiding principle is to corrupt the repeat instance that does *not* contain the masked token—the instance from which the model retrieves the correct amino acid—while leaving the masked instance intact.

We consider four corruption strategies. **BLOSUM** replaces amino acids with their highest-scoring BLOSUM62 substitutions (Henikoff & Henikoff, 1992), preserving biochemical similarity while breaking exact residue identity. **BLOSUM Opposite** replaces amino acids with their lowest-scoring substitutions, introducing strong biochemical dis-similarity that may disrupt the repeat signal more effectively. **Permutation** randomly reorders the repeat segment, preserving composition while destroying order-dependent repetition. **Mask** replaces amino acids with mask tokens, removing residue identity entirely. For the BLOSUM and masking strategies, we corrupt 20%, 50%, and 100% of the non-masked repeat segment. Examples are shown in Table 5.

*Table 5.* **Examples of counterfactual strategies applied to approximate repeats.** Only the repeated segments are shown for brevity. The masked position and corrupted amino acids are highlighted in blue. Underscores indicate masked tokens.

| Counterfactual Type | Input Example |
| --- | --- |
| Original Input | ...RLRHQLSFYG_PFALG...KLRHQLSFYGVAFALG... |
| **100% BLOSUM (Chosen Strategy)** | ...RLRHQLSFYG_PFALG...RIKYEIAYFAISYSIA... |
| Permutation | ...RLRHQLSFYG_PFALG...QLFHYGARFLKLSAVG... |
| 100% BLOSUM Opposite | ...RLRHQLSFYG_PFALG...CDCCCDWPDIRWPWDI... |
| 100% Mask | ...RLRHQLSFYG_PFALG...______________... |
| 50% BLOSUM | ...RLRHQLSFYG_PFALG...KLRHQIAYFGISYALA... |
| 50% BLOSUM Opposite | ...RLRHQLSFYG_PFALG...KLRHQDWPDGRWPALI... |
| 50% Mask | ...RLRHQLSFYG_PFALG...KLRHQ___G__AL_... |
| 20% BLOSUM | ...RLRHQLSFYG_PFALG...KLRHQLSYYGIAFSLG... |
| 20% BLOSUM Opposite | ...RLRHQLSFYG_PFALG...KLRHQLSPYGRAFWLG... |
| 20% Mask | ...RLRHQLSFYG_PFALG...KLRHQLS_YG_AF_LG... |

## G.2 Effect of Counterfactuals on Model Predictions

We evaluate each counterfactual strategy by measuring its effect on ESM-3's predictions across synthetic, identical, and approximate repeat tasks. For each repeat entry, we run ESM-3 on both the clean input (the full sequence) and its counterfactual, measuring differences in top-1 accuracy and the probability assigned to the correct token.

As a baseline, we apply the same corruption to a control segment positioned symmetrically on the opposite side of the masked repeat at the same distance as the non-masked repeat. This positional control helps confirm that effects are specific to repeat corruption rather than general sequence perturbation. For sequences where a valid control segment cannot be constructed due to sequence boundary constraints, the sequence is excluded from baseline evaluation.

Results (Fig. 20) show that all counterfactuals substantially degrade model predictions compared to baseline, confirming that the model relies on the source repeat segment. Full corruption (100% BLOSUM, BLOSUM Opposite, or Mask) produces the strongest degradation. The 50% BLOSUM Opposite is nearly as effective as full corruption while requiring fewer token replacements. Corrupting only 20% has a lower impact. Effects are strongest in synthetic repeats and weaker in natural sequences, suggesting that natural proteins provide additional contextual cues beyond direct repeat matching. Since 100% BLOSUM, 100% BLOSUM Opposite, 100% Mask, Permutation, and 50% BLOSUM Opposite perform comparably, we proceed with multiple counterfactuals for circuit discovery and compare the resulting circuits.

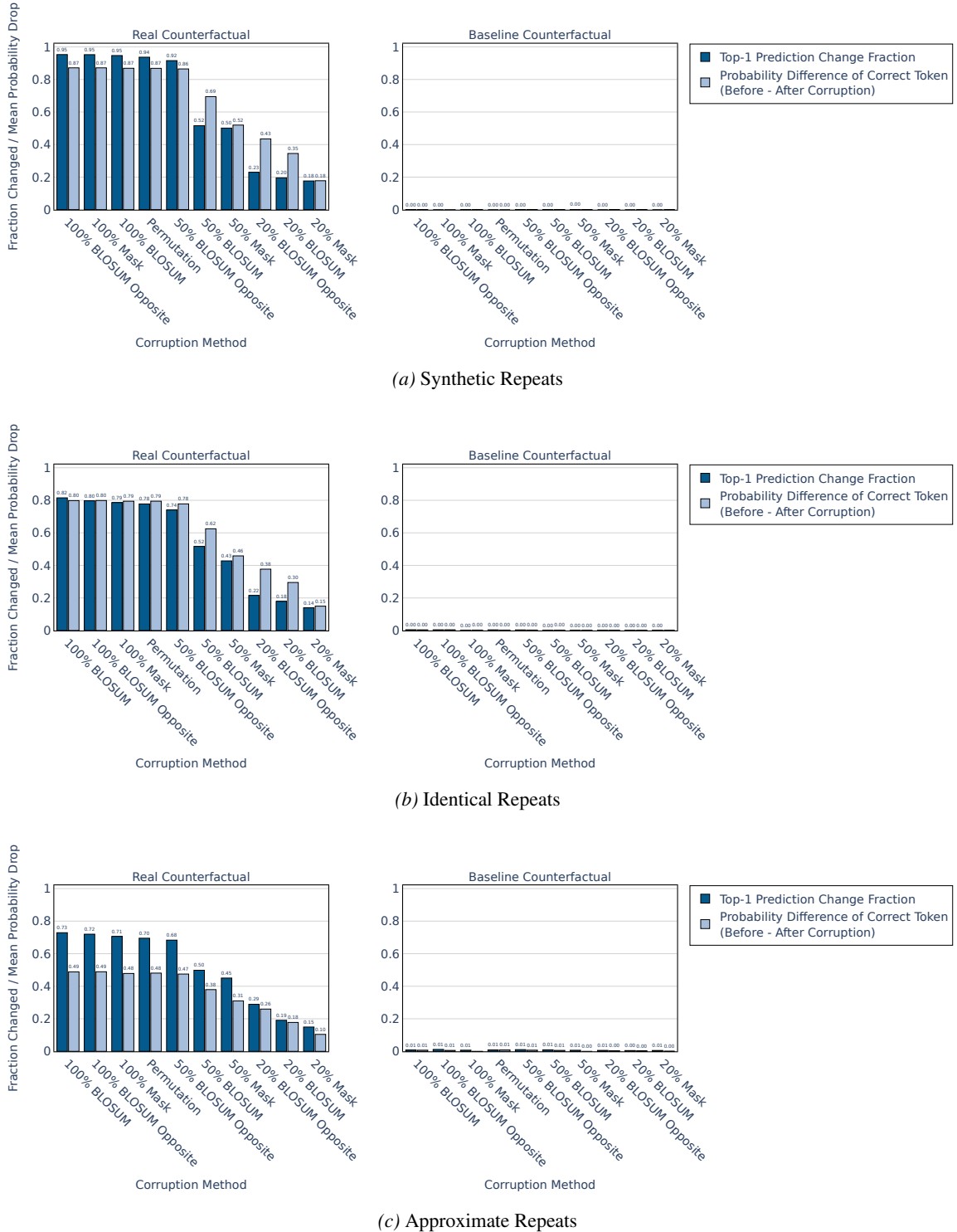

*(a)* Synthetic Repeats

*(b)* Identical Repeats

*(c)* Approximate Repeats

*Figure 20.* Effect of counterfactual corruption strategies on ESM-3 prediction performance across repeat tasks.

## G.3 Circuit Comparison Using Different Counterfactuals

We aim to measure the effect of counterfactual choice on circuit size and overlap, assessing whether different counterfactuals isolate the same repeat identification circuits. We compare four counterfactual strategies—100% BLOSUM, 100% Mask,

50% BLOSUM Opposite, and permutation—by performing circuit discovery on ESM-3 under each strategy.

For each counterfactual strategy and task, we apply circuit discovery and evaluate faithfulness as described in Section 3.1. Results (Fig. 21) show that the 100% BLOSUM strategy yields the smallest circuits at any faithfulness threshold. For synthetic and identical repeats, BLOSUM and BLOSUM Opposite achieve 85% faithfulness at comparable circuit sizes. For approximate repeats, BLOSUM yields the smallest circuits.

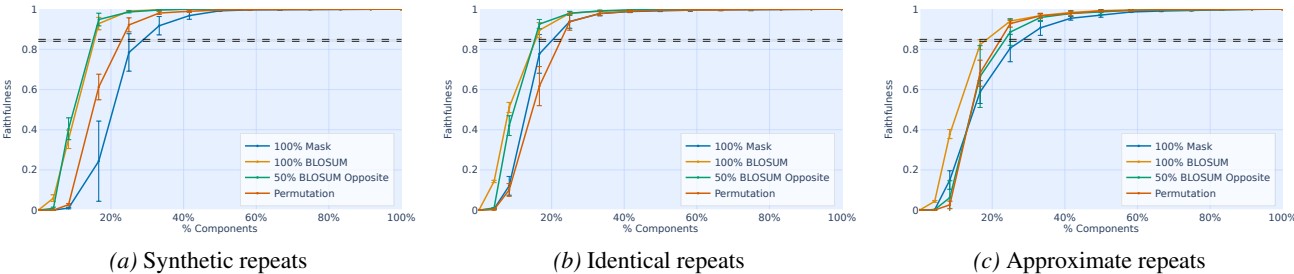

*(a)* Synthetic repeats        *(b)* Identical repeats        *(c)* Approximate repeats

*Figure 21.* **Component-level circuit evaluation for different counterfactual strategies on ESM-3.** Faithfulness is shown as a function of the fraction of included components. Results are averaged across random seeds.

**Cross-counterfactual recall.** We measure circuit overlap by computing cross-counterfactual recall (Fig. 22). Circuits discovered using 100% BLOSUM are largely contained within circuits discovered using other strategies, particularly for approximate repeats. This indicates that different counterfactuals recover a shared core set of components, while some strategies produce larger circuits that include additional components.

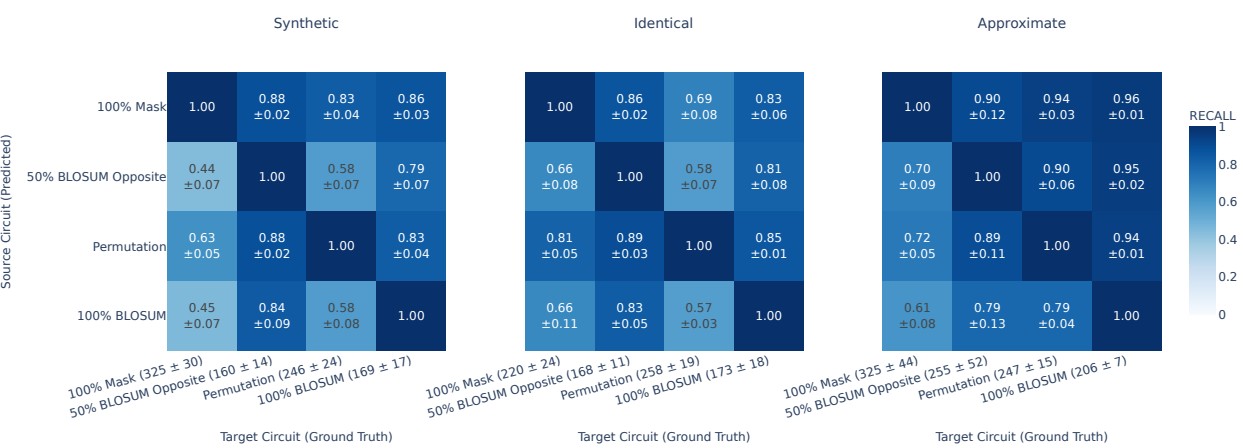

*Figure 22.* **Cross-counterfactual recall of discovered circuits.** Recall measures the fraction of components in the target circuit that are recovered by the source circuit. Circuit sizes (number of components) achieving 85% faithfulness are shown in parentheses, averaged across seeds.

**Cross-counterfactual faithfulness.** To assess functional equivalence across counterfactuals, we evaluate circuits discovered under one counterfactual (at 85% faithfulness) using different counterfactuals for evaluation.

Results (Fig. 23) show that circuits maintain comparable faithfulness across counterfactuals. Even compact circuits discovered with 100% BLOSUM or 50% BLOSUM Opposite retain high faithfulness when evaluated under counterfactuals that typically produce larger circuits, suggesting that some counterfactuals better rank components for repeat identification.

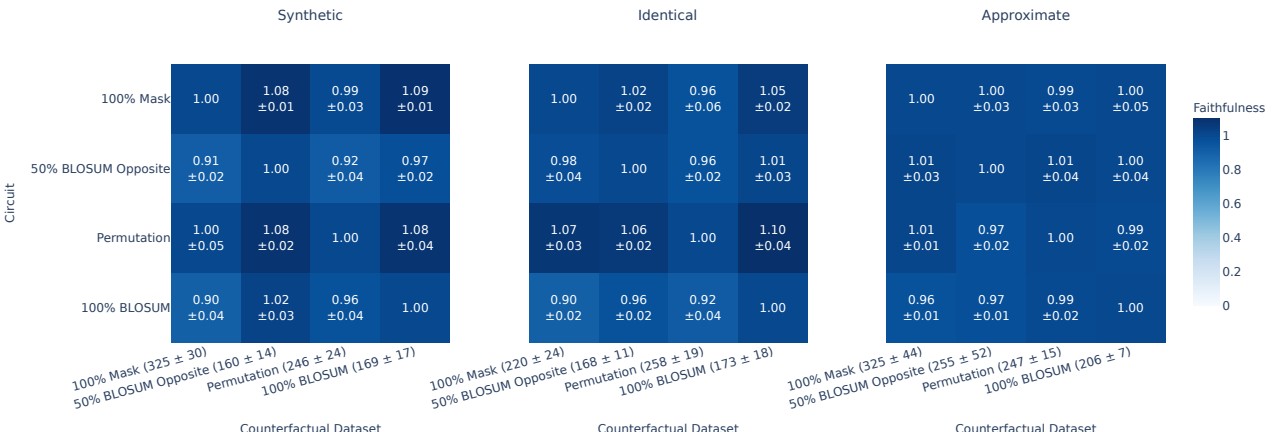

*Figure 23.* **Cross-counterfactual faithfulness of discovered circuits.** Rows correspond to counterfactuals used for circuit discovery, columns correspond to counterfactuals used in evaluation. Faithfulness is normalized per column, relative to the circuit discovered under that counterfactual. Circuit sizes (number of components) achieving $85\%$ faithfulness are shown in parentheses.

**Summary.** Counterfactual choice affects circuit compactness more than the underlying mechanism. While all strategies recover a shared core mechanism, the $100\%$ BLOSUM counterfactual ranks repeat identification components more effectively, yielding more compact circuits. We thus use it as our primary counterfactual throughout this work.

### G.4 Neuron-Level Robustness to Counterfactual Choice

Because our neuron-concept matching includes BLOSUM-based biological concepts, we further test whether the identified neuron-level patterns depend on the BLOSUM counterfactual used during circuit discovery. We repeat the neuron discovery analysis on circuits discovered using the permutation counterfactual for the approximate-repeat task. Of the neurons discovered with the BLOSUM counterfactual, $83.29\%$ are also recovered in the permutation-based circuits; among neurons matched to BLOSUM62 or other biological concepts, $73.5\%$ are rediscovered. These results indicate that the neuron-level patterns identified in our analysis are not specific to the BLOSUM counterfactual.

## H Circuit Discovery and Evaluation: Technical Details

### H.1 Data Filtering

We perform circuit discovery on sequences where the model demonstrates strong task performance. We measure mean model accuracy across all eligible masked positions within the repeating segment. For synthetic and identical repeat settings, we retain only sequences with perfect accuracy. For the approximate repeat setting, we retain only sequences with mean accuracy greater than $80\%$ (See Appendix D for evaluation details). We further retain only sequences for which the counterfactual causes a change in the model's prediction, ensuring a meaningful intervention signal. For each setting (synthetic, identical and approximate repeats) we sample $1,000$ sequences per random seed, using half of them for circuit discovery and the rest of evaluation.

### H.2 Circuit Discovery and Cross-task Comparisons

To identify circuits composed of attention heads and MLPs, we compute importance scores for each component using attribution patching with integrated gradients (AP-IG) with 5 integration steps, following Section 3.1. Importance scores are averaged across 500 input–counterfactual pairs. We rank components by the absolute importance score, and build circuits of progressively increasing sizes ($n_{components} \in \{100, 200, 300, ..., n_{model}\}$, where $n_{model}$ is the number of components in the entire circuit), including the highest-ranking components. To achieve a finer grained circuit that achieves $85\%$ faithfulness, we use binary search over circuit sizes that cross the $85\%$ threshold. We discover one circuit per task per random seed, then compare circuits across task settings using three cross-task metrics (Section 3.2). We repeat the discovery

and evaluation process five times with different random seeds, each time sampling new data subsets and independently discovering and comparing circuits. This verifies that observed similarities and differences between circuits are consistent across data samples. The circuit discovery pipeline was implemented using a fork of TransformerLens (Nanda & Bloom, 2022), which will be made public with the paper.

### H.3 Circuit Neuron-level Analysis

We apply the neuron-level analysis from Section 4.2 to each approximate repeats circuit discovered above, using the same random seed and data subset for consistency. For each circuit, we fix the set of attention heads and MLP layers that were sufficient to achieve approximately $85\%$ circuit faithfulness. We then rank neurons within the selected MLP layers by the absolute value of their importance scores. For each circuit, we fix the set of attention heads and MLP layers that achieve $85\%$ circuit faithfulness, and rank neurons within circuit MLP layers by absolute importance score. We progressively retain a percentage of higher-ranked neurons ($p_{neurons} \in \{0.1\%, 0.2\%, 0.5\%, 1\%, 2\%, 5\%, 10\%, 20\%, 50\%, 100\%\}$, following Mueller et al. (2025)) while ablating all other components: non-circuit attention heads, non-circuit MLP layers, and non-retained neurons within circuit MLP layers. We measure the circuit faithfulness for each such circuit. To preserve most circuit behavior and retain most of the faithfulness, we target approximately $80\%$ faithfulness. We use binary search to find the neuron-retention percentage achieving this threshold.

### H.4 Consistency Across Random Seeds

We focus our analysis on components consistently selected across the five random seeds. We do so, we retain only components that appear in at least 4 out of 5 seed-specific circuits.

**Attention heads and MLPs.** Discovered circuits average $158 \pm 8$ attention heads and $46 \pm 1$ MLPs in ESM-3, and $109 \pm 5$ attention heads and $34 \pm 1$ MLPs in ESM-C. Applying the consistency threshold yields $149/180$ attention heads and $45/47$ MLPs in ESM-3, and $102/124$ attention heads and $32/35$ MLPs in ESM-C, where denominators indicate distinct components appearing in any circuit. This demonstrates high cross-seed stability for attention heads and MLPs.

**Neurons.** The fraction of retained neurons per layer averages $2.8\% \pm 0.3\%$ in ESM-3 and $7.6\% \pm 1.4\%$ in ESM-C. The consistency threshold yields $3394/9461$ neurons in ESM-3 and $3510/19233$ neurons in ESM-C (denominators indicate neurons appearing in any circuit). Neuron-level selection is noisier than head-level selection, particularly in ESM-C. This variability likely reflects our use of fixed retention percentages per layer and the presence of example-specific neurons rather than consistently reused components.

**Validation.** Components that pass the consistency threshold closely align with component ranking by absolute importance score (overlap = $98.9\%$ for ESM-3 and $98.5\%$ for ESM-C at the head/MLP level; $84\%$ for ESM-3 and $80\%$ for ESM-C at the neuron level). This confirms that cross-seed consistency aligns with attribution-based importance.

### H.5 Edge-Level Circuits and Interaction Graph

To quantify relationships between component groups, we employ Edge Attribution Patching with Integrated Gradients (EAP-IG) (Hanna et al., 2024). An edge represents a directed connection between two components (an attention head or an MLP), corresponding to the contribution of a source component's output to a downstream target component's input. EAP-IG assigns each edge an importance score approximating the effect on task performance of replacing the source component's activation with a counterfactual activation. We identify edge-level circuits—connected subsets of edges sufficient for predicting a masked token in the approximate-repeats dataset—by computing EAP-IG scores with 5 integration steps and selecting edges greedily by absolute importance (Hanna et al., 2024). For a given circuit size, we evaluate circuit faithfulness by ablating non-circuit edges with counterfactual activations, and choose the minimal edge amount achieving $85\%$ faithfulness. Repeating this process across 5 random seeds, we find that sufficient circuits require $5.20\% \pm 1.02\%$ of edges in ESM-3 and $3.29\% \pm 0.62\%$ of edges in ESM-C, demonstrating sparsity.

For the analysis of interactions between components groups (Section 5) we consider circuit edges that are consistent across 4 out of 5 random seeds, filter out edges that show both positive and negative importance, and compute the importance score of an edge as the average across seeds. For readability, we scale these aggregated edge scores by a factor of 1000. We aggregate edge scores at the component-group level by averaging over all positive edges connecting a source group to a destination group, yielding the average interactions table value (see Fig. 7 and Fig. 17).

# I Attention Heads Clustering

## I.1 Additional Examples Of Attention Patterns

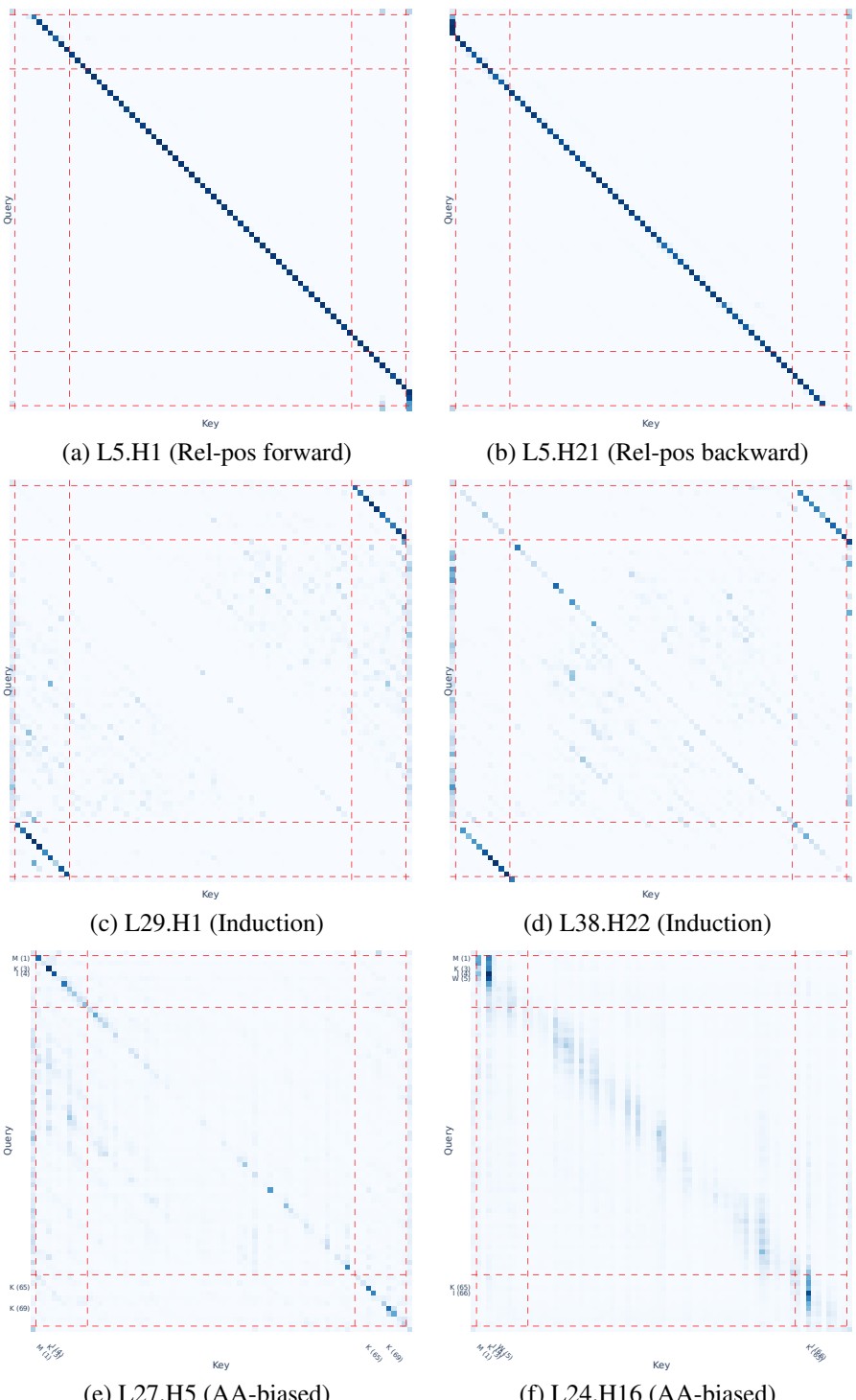

(a) L5.H1 (Rel-pos forward)

(b) L5.H21 (Rel-pos backward)

(c) L29.H1 (Induction)

(d) L38.H22 (Induction)

(e) L27.H5 (AA-biased)

(f) L24.H16 (AA-biased)

*Table 6.* **Example ESM-3 attention patterns on a protein with two repeats** (UniProt A0A2M8A3Y9): MVKIWGREDG (1–10) and MVKIWGKKDG (63–72). Repeated regions are marked by red dashed vertical and horizontal lines. Panels (e,f) show AA-biased heads that activate primarily within repeats, motivating the repeat focus score.

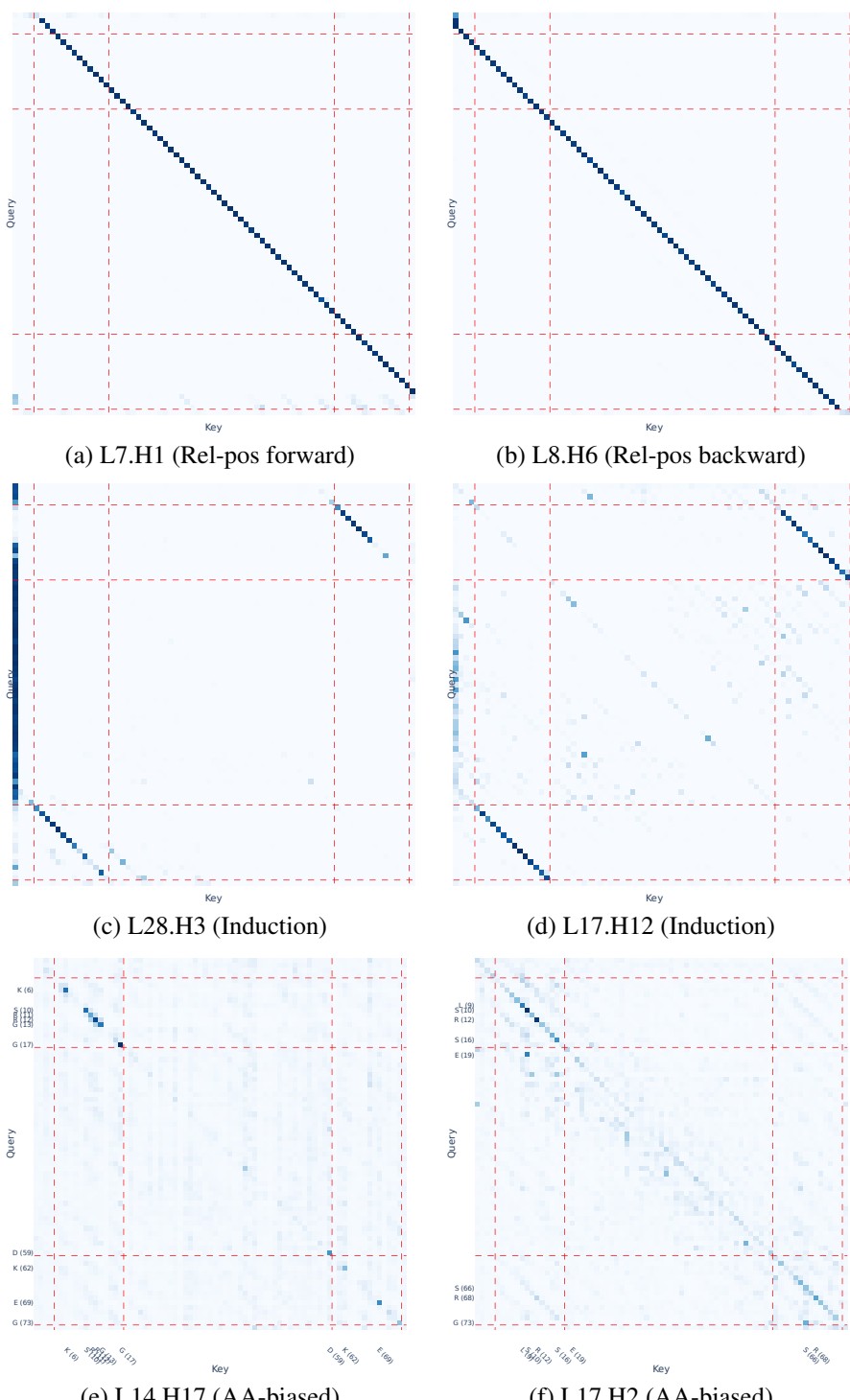

(a) L7.H1 (Rel-pos forward)          (b) L8.H6 (Rel-pos backward)

(c) L28.H3 (Induction)               (d) L17.H12 (Induction)

(e) L14.H17 (AA-biased)              (f) L17.H2 (AA-biased)

*Table 7.* **Example attention patterns from ESM-C on a UniRef50 protein (A0A8X6HTE9_TRICU).** The sequence contains two approximate repeats, TRKGELSRRGCSSG (positions 4–17) and TRKGMLSRRECSSG (positions 60–73). Repeated regions are indicated by red dashed vertical and horizontal lines, marking the start and end of the repeat span along the key (x-axis) and query (y-axis), respectively. (a,b) Heads attending to fixed relative positional offsets. (c,d) Induction-like heads attending between aligned positions across repeated segments. (e,f) Amino-acid–biased heads focusing on specific amino acids within repeated regions, as in ESM-3.

## I.2    Attention Heads Features Description and Computation

In Section 4.1, we presented a high-level description of the attention-head features used for clustering. Here, we provide their formal definitions and exact computation.

We designed a set of quantitative features to characterize each attention head, based on both its attention patterns and its contributions to the residual stream. All attention-head features were computed on the *Approximate Repeats* dataset used in the circuit-discovery experiments which contains approximately 3000 proteins. We excluded a small subset of sequences (16 sequences) in which more than one repeat segment type was eligible for analysis, in order to avoid ambiguity in the metric definitions and to simplify downstream aggregation.

We introduce the following notation, which is used to formally define the attention head features in Table 8.

- $\mathcal{V}$: the vocabulary of the protein language model (PLM). For ESM-3, $\mathcal{V}$ refers to the sequence-token vocabulary (as multiple vocabularies exist, one per track).

- $s$: a protein sequence of length $L$, represented in tokenized form as

$$s = (t_0, t_1, \ldots, t_{L+1}),$$

  where $t_0 = \text{BOS}$, $t_{L+1} = \text{EOS}$, and each $t_i \in \mathcal{V}$ for $1 \leq i \leq L$ is an amino-acid token. In our task setup, exactly one position among $\{1, \ldots, L\}$ is replaced with the mask token.

- $\mathcal{U}(s)$: the set of unique tokens appearing in sequence $s$,

$$\mathcal{U}(s) = \{\, t_i \mid 0 \leq i \leq L+1 \,\} \subseteq \mathcal{V}.$$

- $\mathcal{R}(s)$: the set of positions corresponding to repeat tokens in sequence $s$.

- $\mathcal{R}_{\text{pairs}}(s)$: the set of aligned repeat-position pairs in sequence $s$. Each element is a pair $(q, k)$ such that token $t_q$ in one repeat instance is aligned with token $t_k$ in the other instance. The set is symmetric: if $(q, k) \in \mathcal{R}_{\text{pairs}}(s)$, then $(k, q) \in \mathcal{R}_{\text{pairs}}(s)$.

- $\mathcal{N}(s)$: the set of non-repeat token positions in $s$, containing all valid sequence-token positions that are not part of either repeat instance. Special tokens BOS (position 0) and EOS (position $L+1$) are excluded.

- $A_h^{(s)}$: the attention pattern of head $h$ for sequence $s$,

$$A_h^{(s)} \in \mathbb{R}^{(L+2) \times (L+2)},$$

  where rows correspond to query positions and columns to key positions, and $A_h^{(s)}[q, k]$ denotes the attention weight from query $q$ to key $k$.

- $m(s)$: the (unique) masked position in sequence $s$.

- $P_{h,q}^{(s)}$: the vocabulary distribution induced by head $h$ at token position $q$ in sequence $s$. Let $V_{h,q}^{(s)} \in \mathbb{R}^{|\mathcal{V}|}$ be the logit vector obtained by applying the Logit Lens to the output of head $h$ at position $q$. Then, for $t \in \mathcal{V}$,

$$P_{h,q}^{(s)}(t) = \text{softmax}\big(V_{h,q}^{(s)}\big)_t,$$

  where $P_{h,q}^{(s)}(t)$ denotes the probability assigned to token $t$.

- $\widetilde{P}_{h,q}^{(s)}$: the normalized vocabulary distribution induced by head $h$ at token position $q$,

$$\widetilde{P}_{h,q}^{(s)}(t) = \max\left( P_{h,q}^{(s)}(t) - \tfrac{1}{|\mathcal{V}|} \sum_{t' \in \mathcal{V}} P_{h,q}^{(s)}(t'),\ 0 \right),$$

  which highlights probabilities elevated relative to the vocabulary mean.

- $K_a(s)$: the set of positions in $s$ where amino acid $a$ occurs,

$$K_a(s) = \{\, k \in \{1, \ldots, L\} : t_k = a \,\}.$$

- $\mathcal{AA}$: the set of the 20 standard amino acids.

- $\mathbf{o}_{h,q}^{(s)}$: the output vector of attention head $h$ at query position $q$ in sequence $s$.

- $\Delta \mathbf{r}_q^{(s)}$: the residual stream update at token position $q$ induced by the attention sublayer in the transformer layer containing head $h$,

$$\Delta \mathbf{r}_q^{(s)} = \mathbf{r}_q^{\text{post-attn}} - \mathbf{r}_q^{\text{pre}},$$

where $\mathbf{r}_q^{\text{pre}}$ denotes the residual stream input to the attention sublayer and $\mathbf{r}_q^{\text{post-attn}}$ its output.

- $\mathcal{D}$: the dataset of protein sequences used to compute attention-head features. Each element $s \in \mathcal{D}$ is a tokenized protein sequence with exactly one masked.

*Table 8.* Summary of attention head features computed per protein sequence

| Feature | Formal Definition | Description |
|---|---|---|
| Induction Score | $\text{Induction}_h = \frac{1}{2}\left(\text{Copying}_h + \text{PM}_h\right)$ 

 $\text{Copying}_h = \frac{1}{\lvert\mathcal{D}\rvert} \sum_{s\in\mathcal{D}} \left( \frac{\widetilde{P}_{h,m(s)}^{(s)}\left(t_{j^*(s)}\right)}{\sum_{t\in\mathcal{U}(s)} \widetilde{P}_{h,m(s)}^{(s)}(t)} \right)$ 
 where $j^*(s) = \arg\max_{j\in\{0,\ldots,L+1\}} A_h^{(s)}[m(s), j]$ 

 $\text{PM}_h = $ 
 $\frac{1}{\lvert\mathcal{D}\rvert} \sum_{s\in\mathcal{D}} \left( \frac{1}{\lvert\mathcal{R}_{\text{pairs}}(s)\rvert} \sum_{(q,k)\in\mathcal{R}_{\text{pairs}}(s)} A_h^{(s)}[q,k] \right)$ | Measures induction-like behavior following prior work on decoder-only models (Olsson et al., 2022; Ren et al., 2024; Bansal et al., 2023), which defines induction behavior as the combination of pattern-matching and copying. The Copying score quantifies whether the head promotes the token it attends to most strongly at the masked position, while the Pattern Matching component measures attention between aligned tokens across the two repeat instances. Both scores are first computed at the per-sequence level and averaged across all sequences. The Induction Score is then defined as the mean of these two aggregated quantities, yielding a single score per head. This score is motivated by manual inspection of attention patterns that closely resemble induction behavior previously identified in large language models. |

**Table 8 (continued)**

| Feature | Formal Definition | Description |
|---|---|---|
| Relative Position Score | $X_q^{(s)} = \frac{(\arg\max_k A_h^{(s)}[q,k]) - q}{L}$ 
 $\mu_h^{(s)} = \frac{1}{L} \sum_{q=1}^{L} X_q^{(s)}$ 
 $\mathrm{RP}_h^{(s)} = \sqrt{\frac{1}{L} \sum_{q=1}^{L} \left( X_q^{(s)} - \mu_h^{(s)} \right)^2}$ 
 $\mathrm{RP}_h = 1 - \frac{1}{|\mathcal{D}|} \sum_{s \in \mathcal{D}} \mathrm{RP}_h^{(s)}$ | For each sequence $s$ and query position $q$, $X_q^{(s)}$ denotes the normalized relative offset between the query and the key position receiving maximal attention from head $h$. The per-sequence score $\mathrm{RP}_h^{(s)}$ is computed as the standard deviation of $X_q^{(s)}$ across query tokens, with $\mu_h^{(s)}$ denoting the mean offset. Offsets are normalized by the sequence length $L$ to allow comparison across sequences of different lengths, and BOS/EOS positions are excluded as queries. The final score $\mathrm{RP}_h$ is obtained by averaging $\mathrm{RP}_h^{(s)}$ over the dataset $\mathcal{D}$ and inverting it $(1 - \cdot)$, so that higher values indicate heads that attend to a more consistent relative position. This metric is motivated by qualitative observations of heads attending to fixed relative offsets (e.g., $\pm n$ tokens). |
| Attn. Amino-Acid Bias | $\mathrm{AA\_Bias}_{h,a}^{(s)} = \frac{1}{L} \sum_{q=1}^{L} \sum_{k \in K_a(s)} A_h^{(s)}[q,k]$ 
 $\mathrm{AA\_Bias}_{h,a} = \frac{1}{|\mathcal{D}|} \sum_{s \in \mathcal{D}} \mathrm{AA\_Bias}_{h,a}^{(s)}$ 
 $P_h(a) = \frac{\mathrm{AA\_Bias}_{h,a}}{\sum_{a' \in \mathcal{A}\mathcal{A}} \mathrm{AA\_Bias}_{h,a'}}$ 
 $\mathrm{AA\_Score}_h = \mathrm{JSD}(P_h \parallel P_{\mathrm{data}})$ | For each amino acid $a$, we compute the average attention mass that head $h$ assigns from sequence-token queries to all key positions containing $a$, and average this quantity across the dataset, yielding one feature per amino acid. These features are normalized to form an amino-acid attention distribution for each head, which is compared to the dataset amino-acid frequency distribution using Jensen–Shannon divergence. This score is motivated by qualitative observations of attention patterns, including global amino-acid biases, where a head broadly attends to all occurrences of a subset of amino acids, as well as more localized, motif-like patterns, where attention is concentrated on specific amino acids at particular positions. |

**Table 8 (continued)**

| Feature | Formal Definition | Description |
|---|---|---|
| Repeat Focus | $\mathrm{RF}_h^{(s)} = \log \frac{\frac{1}{|\mathcal{R}(s)|} \sum_{q \in \mathcal{R}(s)} \max_k A_h^{(s)}[q,k]}{\frac{1}{|\mathcal{N}(s)|} \sum_{q \in \mathcal{N}(s)} \max_k A_h^{(s)}[q,k]}$ 
 $\mathrm{RF}_h = \frac{1}{|\mathcal{D}|} \sum_{s \in \mathcal{D}} \mathrm{RF}_h^{(s)}$ | Measures how differently head $h$ behaves when the *query* comes from a repeat segment compared to non-repeat regions, by partitioning query positions into repeat queries $\mathcal{R}(s)$ and non-repeat queries $\mathcal{N}(s)$. For each query, the maximum attention peak across keys is computed and averaged separately over the two sets. The logarithm of their ratio centers the measure at zero: positive values indicate sharper focus on repeat queries, negative values indicate sharper focus on non-repeat queries, and values near zero indicate similar behavior across regions. The score is computed per sequence and averaged across all sequences in the dataset. This score is motivated by manual observations that, beyond induction heads, additional heads tend to exhibit stronger activation for repeat queries. |
| Attention to BOS/EOS | $\mathrm{A}_h^{\mathrm{BOS}} = \frac{1}{|\mathcal{D}|} \sum_{s \in \mathcal{D}} \frac{1}{L} \sum_{q=1}^{L} A_h^{(s)}[q, 0]$ 
 $\mathrm{A}_h^{\mathrm{EOS}} = \frac{1}{|\mathcal{D}|} \sum_{s \in \mathcal{D}} \frac{1}{L} \sum_{q=1}^{L} A_h^{(s)}[q, L+1]$ 
 $\mathrm{A}_h^{\mathrm{BOS/EOS}} = \frac{1}{2} \left( \mathrm{A}_h^{\mathrm{BOS}} + \mathrm{A}_h^{\mathrm{EOS}} \right)$ | Measures the average attention weight that head $h$ assigns from sequence-token query positions to the sequence-boundary positions corresponding to the beginning of the sequence (BOS) and the end of the sequence (EOS). Throughout, positions $0$ and $L + 1$ denote the BOS and EOS boundary positions, respectively. Both scores are computed at the per-example level and averaged across all sequences. This metric is motivated by manual inspection of attention patterns, where we observed several heads that attend to boundary positions in a subset of proteins examined qualitatively. Prior work suggests that attention to sequence-boundary positions can be associated with "no-op" behavior or approximate if–else mechanisms, in which heads default to boundary positions when no salient pattern is detected (Clark et al., 2019; Vig & Belinkov, 2019; Barbero et al., 2025). |

**Table 8 (continued)**

| Feature | Formal Definition | Description |
|---|---|---|
| Attention Entropy | $\text{ENT}_h^{(s)} = \frac{1}{L} \sum_{q=1}^{L} \frac{-\sum_{k=0}^{L+1} A_h^{(s)}[q,k] \log A_h^{(s)}[q,k]}{\log(L+2)}$ 
 $\text{ENT}_h = \frac{1}{|\mathcal{D}|} \sum_{s \in \mathcal{D}} \text{ENT}_h^{(s)}$ | Computes the average normalized Shannon entropy of the attention distribution of head $h$ over sequence-token queries, excluding BOS/EOS as queries but including them as keys. For each query, entropy is normalized by $\log(L + 2)$, corresponding to the tokenized sequence length, yielding values in $[0, 1]$ and enabling comparison across sequences of different lengths. The score is averaged across all sequences in the dataset. High entropy indicates that the head spreads attention broadly across many tokens, while low entropy indicates that the head focuses attention sharply on a few tokens. This metric is motivated by empirical observations that relative-position and induction heads typically exhibit lower entropy than other heads, making attention entropy useful for separating distinct head behaviors. |
| Context Sensitivity | $\text{KL}_h^{(s)} =$ 
 $\frac{1}{L+2} \sum_{q,k} A_h^{\text{original},(s)}[q,k] \log \frac{A_h^{\text{original},(s)}[q,k]}{A_h^{\text{counterfactual},(s)}[q,k]}$ 
 where $q, k \in \{0, \dots, L+1\}$. 
 $\text{KL}_h = \frac{1}{|\mathcal{D}|} \sum_{s \in \mathcal{D}} \text{KL}_h^{(s)}$ | Computes the average Kullback–Leibler divergence between the attention distribution of head $h$ under the original input and under a counterfactual input, computed per sequence. For each sequence $s$, the divergence is evaluated for each query token by comparing its attention distribution over keys in the original sequence to that induced by a counterfactual sequence, and then averaged across queries. High values indicate that the head is sensitive to sequence context and substantially changes its attention in response to the disruption, whereas low values indicate stable, context-invariant attention patterns (e.g., fixed-offset heads). This score helps distinguish amino-acid–biased heads from fixed-position heads: although some amino-acid–biased heads can exhibit localized attention and attain high relative-position scores, they typically lack the consistent, token-invariant patterns characteristic of fixed-offset position heads. |

**Table 8 (continued)**

| Feature | Formal Definition | Description |
|---|---|---|
| Relative Contribution to Residual Stream | $\text{CRS}_h^{(s)} = \frac{1}{|\mathcal{R}(s)|} \sum_{q \in \mathcal{R}(s)} \frac{\|\mathbf{o}_{h,q}^{(s)}\|_2}{\|\Delta \mathbf{r}_q^{(s)}\|_2}$ $\text{CRS}_h = \frac{1}{|\mathcal{D}|} \sum_{s \in \mathcal{D}} \text{CRS}_h^{(s)}$ | Computes the average ratio between the norm of the output vector $\mathbf{o}_{h,q}$ produced by head $h$ and the norm of the full residual update $\Delta \mathbf{r}_q$, evaluated at repeat query positions $q \in \mathcal{R}(s)$. The score is computed per sequence and then averaged across the dataset. Higher values indicate that the head contributes more strongly to the attention-layer residual stream update relative to other heads in the same layer. This metric, introduced in (Reusch & Belinkov, 2025), is intended to capture the relative importance of attention heads in shaping the residual stream at the layer they are in. |
| Vocabulary Entropy | $\text{VocabENT}_h^{(s)} = \frac{-\sum_{t \in \mathcal{V}} P_h^{\text{mask}}(s)(t) \log P_h^{\text{mask}}(s)(t)}{\log |\mathcal{V}|}$ $\text{VocabENT}_h = \frac{1}{|\mathcal{D}|} \sum_{s \in \mathcal{D}} \text{VocabENT}_h^{(s)}$ | Computes the normalized Shannon entropy of the vocabulary distribution induced by head $h$ at the masked token position. The distribution $P_h^{\text{mask}}(s)$ is obtained by applying the Logit Lens to the output of head $h$ at the masked position in sequence $s$. The entropy is normalized by $\log |\mathcal{V}|$, where $|\mathcal{V}|$ is the vocabulary size, yielding values in $[0, 1]$. This metric aims to capture whether a head has a stronger effect on the final prediction. Lower values indicate that the head concentrates probability mass on a small set of tokens and thus exerts a stronger, more selective influence on the final prediction. |

### I.3   Clustering Details

To group attention heads into functionally similar categories, we applied *k*-means clustering (Lloyd, 1982; Arthur & Vassilvitskii, 2007) to the standardized attention-head feature matrix using the `scikit-learn` (Pedregosa et al., 2011) implementation. All features were standardized to zero mean and unit variance prior to clustering.

To select the number of clusters, we evaluated both the Elbow method (based on inertia) and the Silhouette score (Rousseeuw, 1987) for $k \in [2, 10]$, separately for ESM-3 and ESM-C. As shown in Figure 24, the inertia curves for both models exhibit an elbow at $k = 3$, indicating that increasing the number of clusters beyond three yields only minor improvements in within-cluster compactness.

The Silhouette analysis yields different behaviors across models. For ESM-3 (Figure 24, top row), the Silhouette score is maximized at $k = 3$ and decreases for larger values of $k$, supporting the choice suggested by the inertia criterion. In contrast, for ESM-C (Figure 24, bottom row), the Silhouette score is more ambiguous: while the maximum is attained at $k = 2$, the score at $k = 3$ remains comparable and does not exhibit a sharp drop.

Given this ambiguity, we additionally considered qualitative inspection of the resulting clusters. Empirically, using $k = 3$ consistently separates amino-acid–biased heads from fixed-position heads, whereas clustering with $k = 2$ tends to merge these two functionally distinct groups. Based on the clear elbow observed in the inertia curves for both models and the

improved functional interpretability of the clusters, we selected $k = 3$ as the number of clusters for both ESM-3 and ESM-C. We further verified in Appendix I.4.2, for both models, that the clusters remain distinct using a statistical test.

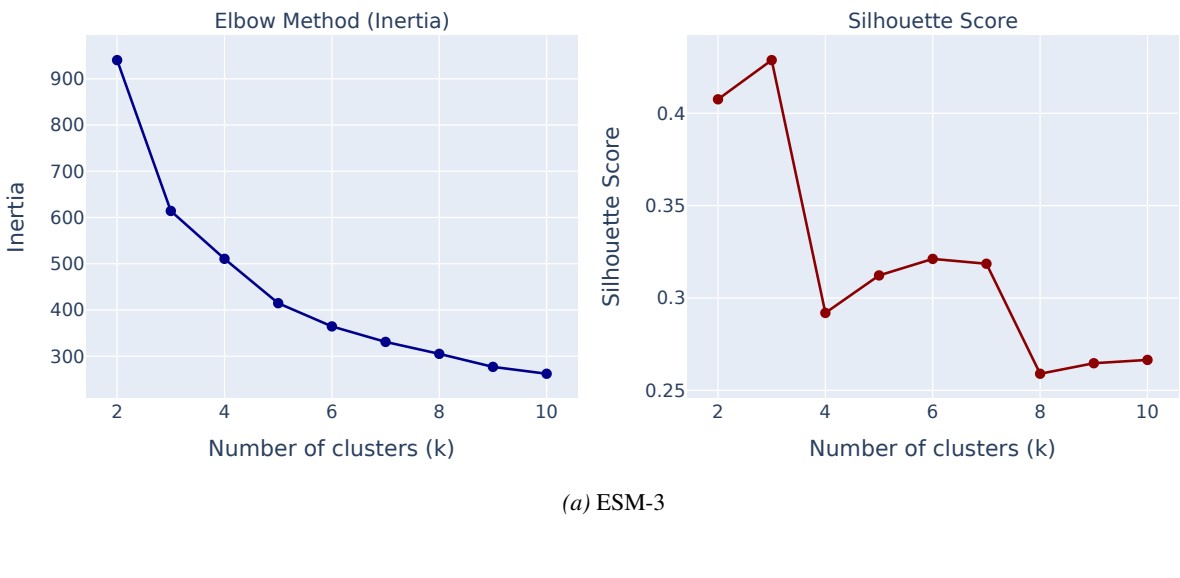

*(a)* ESM-3

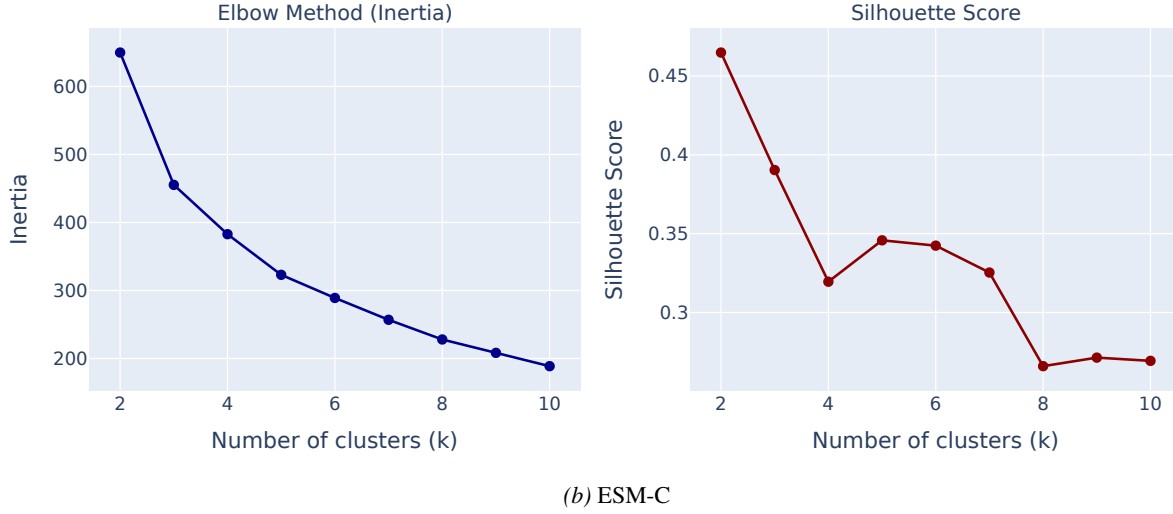

*(b)* ESM-C

*Figure 24.* Cluster selection metrics for attention-head clustering. We report inertia (Elbow method, left) and Silhouette scores (right) for $k \in [2, 10]$ for ESM-3 (top) and ESM-C (bottom).

## I.4 Clustering Results

### I.4.1 Cluster Assignments

*Table 9.* Cluster assignments for $k = 3$ obtained from $k$-means clustering over attention-head features for ESM-3.

| Cluster | Heads |
|---|---|
| $C_0$ | A0.H7, A0.H17, A3.H9, A3.H12, A3.H14, A5.H1, A5.H14, A5.H21, A7.H19, A7.H21, A8.H9, A9.H15, A9.H17, A15.H10, A22.H2, A22.H9, A22.H10, A22.H13, A24.H9, A25.H13, A25.H14, A25.H16, A26.H0, A26.H2, A27.H7, A27.H11, A27.H12, A27.H13, A28.H10, A28.H11, A30.H2, A30.H22, A30.H23, A31.H9, A32.H4, A32.H14, A32.H21, A34.H23, A35.H1, A35.H3, A35.H20, A36.H2, A36.H3, A36.H4, A36.H9, A36.H10, A36.H15, A36.H20, A37.H0, A37.H2, A37.H4, A37.H9, A37.H19, A39.H7, A39.H8, A39.H13, A39.H15, A39.H17, A39.H22, A40.H1, A40.H8, A40.H13, A40.H19, A40.H21, A41.H2, A41.H3, A41.H7, A41.H8, A41.H14, A41.H20, A43.H2, A43.H3, A43.H4, A44.H1, A44.H2, A44.H9, A44.H11, A44.H16, A44.H18, A45.H0, A45.H3, A45.H4, A45.H17, A45.H23, A46.H1, A46.H2, A46.H3, A46.H4, A46.H8, A46.H16, A47.H5, A47.H8, A47.H10, A47.H12, A47.H14, A47.H15, A47.H17, A47.H21, A47.H22 |
| $C_1$ | A23.H5, A23.H23, A28.H13, A29.H1, A38.H10, A38.H22, A42.H5 |
| $C_2$ | A0.H4, A24.H16, A24.H19, A25.H23, A26.H19, A27.H5, A31.H8, A31.H17, A33.H23, A34.H9, A35.H10, A35.H21, A37.H8, A37.H10, A37.H15, A38.H0, A38.H18, A39.H3, A39.H6, A39.H10, A40.H7, A40.H11, A40.H16, A40.H17, A40.H22, A41.H13, A41.H17, A42.H10, A45.H2, A45.H6, A45.H9, A45.H10, A45.H16, A46.H10, A46.H17, A46.H22, A47.H2, A47.H3, A47.H6, A47.H7, A47.H18, A47.H19, A47.H20 |

*Table 10.* Cluster assignments for $k = 3$ obtained from $k$-means clustering over attention-head features for ESM-C.

| Cluster | Heads |
|---|---|
| $C_0$ | A0.H7, A0.H15, A0.H16, A0.H17, A1.H3, A1.H10, A7.H1, A7.H7, A8.H6, A8.H16, A9.H3, A9.H9, A10.H7, A10.H8, A11.H8, A14.H5, A15.H0, A15.H2, A15.H9, A15.H10, A15.H15, A15.H17, A16.H2, A16.H5, A16.H12, A16.H13, A17.H0, A18.H2, A18.H6, A18.H8, A18.H9, A18.H13, A18.H14, A18.H17, A19.H7, A20.H1, A20.H6, A20.H13, A21.H6, A22.H8, A22.H10, A22.H12, A23.H13, A23.H15, A24.H0, A24.H1, A24.H2, A24.H3, A24.H10, A25.H16, A26.H1, A27.H1, A27.H16, A29.H12, A29.H15, A34.H0, A34.H9, A35.H0, A35.H3, A35.H10, A35.H11, A35.H12 |
| $C_1$ | A12.H2, A13.H5, A17.H4, A17.H12, A19.H8, A19.H14, A19.H16, A21.H0, A28.H3, A29.H13, A30.H12, A35.H16 |
| $C_2$ | A4.H14, A13.H12, A14.H17, A16.H3, A16.H11, A17.H2, A17.H8, A18.H16, A20.H7, A21.H14, A25.H12, A26.H2, A26.H16, A27.H0, A28.H17, A29.H5, A29.H10, A30.H15, A33.H10, A34.H8, A34.H11, A34.H15, A35.H1, A35.H2, A35.H5, A35.H13, A35.H15, A35.H17 |

### I.4.2 Clustering Characterization

In Section 4.1, we note that each cluster aligns with one of the attention-head concepts identified through manual analysis. This section provides a formal characterization of this correspondence based on the cluster-level feature profiles.

To characterize each cluster obtained for ESM-3 and ESM-C, we compute the average value of each attention-head feature used for clustering, aggregated over all heads assigned to the same cluster. Results are reported in Table 11 and Table 12. We further verified these profile distinctions using a one-way Welch ANOVA, applied separately to each attention-head feature, which confirmed that for every reported feature, at least one cluster mean differs significantly across clusters, relative to within-cluster variance, in both ESM-3 and ESM-C ($p < 0.01$).

Across both models, the clustering results closely align with the behavioral distinctions identified during manual inspection,

yielding three interpretable groups of attention heads. Below, we summarize the defining feature profiles of each cluster.

**Cluster 0 (Relative-Position Heads).**   Cluster 0 corresponds to relative-position heads and is characterized by the highest *Relative Position* scores, together with low *Attention to BOS/EOS*, low *Context Sensitivity*, and low *Attention Entropy*, indicating position-specific and context-invariant attention.

**Cluster 1 (Induction Heads).**   Cluster 1 corresponds to induction heads and exhibits the highest *Induction* and *Repeat Focus* scores. These heads also show high *Context Sensitivity* and elevated *Attention to BOS/EOS*, suggesting sensitivity to disruptions in repeat structure. In terms of output contribution, they exhibit the highest *Residual Contribution* together with the lowest *Vocabulary Entropy*, indicating a strong influence on the final prediction.

**Cluster 2 (Amino-Acid–Biased Heads).**   Cluster 2 corresponds to amino-acid–biased (AA-biased) heads and exhibits the highest *Amino Acid Bias*, together with moderate *Relative Position* and *Context Sensitivity*, indicating attention patterns that are context-dependent rather than fixed, in contrast to relative-position heads. While the average *Repeat Focus* is low, a subset of heads exhibits repeat-related attention patterns (Section I.5).

*Table 11.* Feature profiles of attention-head clusters in ESM-3. Values are reported before standardization.

| CLUSTER | SIZE | INDUCTION | REL. POS. | AA BIAS | ATTN. BOS/EOS | ENTROPY | RESIDUAL CONTR. | VOCAB. ENTROPY | CONTEXT SENS. | REPEAT FOCUS |
|---|---|---|---|---|---|---|---|---|---|---|
| $C_0$ | 99 | 0.026 | 0.992 | 0.043 | 0.006 | 0.324 | 0.288 | 0.478 | 0.080 | $-0.002$ |
| $C_1$ | 7 | 0.650 | 0.667 | 0.041 | 0.051 | 0.537 | 0.462 | 0.15 | 0.761 | 1.003 |
| $C_2$ | 43 | 0.045 | 0.817 | 0.128 | 0.043 | 0.765 | 0.193 | 0.564 | 0.283 | 0.108 |

*Table 12.* Feature profiles of attention-head clusters in ESM-C. Values are reported before standardization.

| CLUSTER | SIZE | INDUCTION | REL. POS. | AA BIAS | ATTN. BOS/EOS | ENTROPY | RESIDUAL CONTR. | VOCAB. ENTROPY | CONTEXT SENS. | REPEAT FOCUS |
|---|---|---|---|---|---|---|---|---|---|---|
| $C_0$ | 62 | 0.026 | 0.987 | 0.035 | 0.008 | 0.306 | 0.297 | 0.602 | 0.086 | 0.051 |
| $C_1$ | 12 | 0.493 | 0.696 | 0.040 | 0.072 | 0.503 | 0.322 | 0.445 | 0.759 | 0.857 |
| $C_2$ | 28 | 0.041 | 0.751 | 0.083 | 0.076 | 0.706 | 0.215 | 0.650 | 0.292 | 0.084 |

## I.5   Heads with Repeat Focus

In Section 4.1, we introduce the *Repeat Focus* score to quantify sharper attention to repeat regions. This score is driven by manual observations of attention patterns in which queries corresponding to repeated regions exhibit more concentrated attention (see Section I.1). While induction heads dominate the highest repeat-focus scores, we also observe additional heads outside the induction group that exhibit repeat-focused attention patterns. In this subsection, we identify such heads quantitatively and compare their prevalence across models.

To identify such cases, we analyze the *Repeat Focus* feature and flag heads with unusually high values using an interquartile range (IQR) criterion, specifically score $> Q_3 + 1.5 \cdot$ IQR (Tukey, 1977).

We observe both induction heads and heads from Cluster $C_2$ (amino-acid–biased heads), and therefore report results only for the latter. Focusing on non-induction heads, we find that in ESM-3 a substantial subset of amino-acid–biased heads exhibit elevated repeat focus, whereas in ESM-C only a small number of Cluster $C_2$ heads meet the same criterion. This difference suggests that repeat sensitivity among non-induction heads is more prevalent in ESM-3 than in ESM-C. One plausible contributing factor is the difference in training objectives and supervision: ESM-3 was trained both on sequence data and with additional functional annotation supervision, including annotations associated with known repeat patterns, which may increase sensitivity to biologically meaningful sequence patterns within repeat regions.

*Table 13.* **Non-induction attention heads (Cluster $C_2$) with elevated repeat focus.** Shown are attention heads from Cluster $C_2$ with outlier values of the *Repeat Focus* feature, identified using an IQR-based threshold (score $> Q_3 + 1.5 \cdot \text{IQR}$), for both ESM-3 and ESM-C. Note that, based on the score definition, values above zero indicate that query tokens in repeated regions tend to assign stronger attention to specific key positions than non-repeated region queries, with higher magnitude indicating a stronger repeat-focus phenomenon.

| MODEL | HEAD | REPEAT FOCUS | CLUSTER |
|-------|------|--------------|---------|
| ESM-3 | A26.H19 | 0.701 | $C_2$ |
| ESM-3 | A24.H16 | 0.628 | $C_2$ |
| ESM-3 | A27.H5 | 0.618 | $C_2$ |
| ESM-3 | A46.H22 | 0.565 | $C_2$ |
| ESM-3 | A31.H8 | 0.562 | $C_2$ |
| ESM-3 | A38.H0 | 0.541 | $C_2$ |
| ESM-3 | A47.H2 | 0.463 | $C_2$ |
| ESM-3 | A33.H23 | 0.418 | $C_2$ |
| ESM-3 | A24.H19 | 0.356 | $C_2$ |
| ESM-3 | A35.H21 | 0.334 | $C_2$ |
| ESM-3 | A35.H10 | 0.312 | $C_2$ |
| ESM-3 | A40.H16 | 0.292 | $C_2$ |
| ESM-3 | A45.H6 | 0.279 | $C_2$ |
| ESM-C | A14.H17 | 1.238 | $C_2$ |
| ESM-C | A17.H2 | 0.692 | $C_2$ |
| ESM-C | A20.H7 | 0.588 | $C_2$ |

# J  Neuron Analysis

## J.1  Definition of MLP Neurons

Following Geva et al. (2021), we define an MLP neuron in layer $l$ as a single scalar component of the post-activation vector $h_{\text{post}}^{(l)} \in \mathbb{R}^{d_{\text{mlp}}}$. The MLP output is given by

$$h_{\text{out}}^{(l)} = h_{\text{post}}^{(l)} W_{\text{out}}^{(l)}, \tag{2}$$

where $W_{\text{out}}^{(l)} \in \mathbb{R}^{d_{\text{mlp}} \times d_{\text{model}}}$.

Let $h_{\text{post}}^{(l,i)}$ denote the activation of neuron $i$ in layer $l$, and $v_{\text{out}}^{(l,i)}$ the corresponding row of $W_{\text{out}}^{(l)}$. The MLP output can therefore be written as:

$$h_{\text{out}}^{(l)} = \sum_{i=1}^{d_{\text{mlp}}} h_{\text{post}}^{(l,i)} v_{\text{out}}^{(l,i)}.$$

ESM-3 and ESM-C use gated MLPs (Shazeer, 2020), where the post-activation is computed as:

$$h_{\text{post}}^{(l)} = \sigma\left(h_{\text{in}}^{(l)} W_1^{(l)}\right) \odot \left(h_{\text{in}}^{(l)} W_2^{(l)}\right),$$

where $h_{\text{in}}^{(l)} \in \mathbb{R}^{d_{\text{model}}}$, $W_1^{(l)}, W_2^{(l)} \in \mathbb{R}^{d_{\text{model}} \times d_{\text{mlp}}}$, and $\odot$ denotes element-wise multiplication.

## J.2  Neuron Concept Definitions

This section provides the full list of neuron concepts used to systematically characterize neurons, as described in Section 4.2, and is driven by manual observations. The concept set focuses on grouping amino acids according to biologically plausible shared properties, reflecting our observation that many neurons activate selectively on specific subsets of amino acids. While the space of possible amino-acid subsets is combinatorially large ($2^{20}$), we restrict our attention to a curated set of concepts that reflect established biochemical, structural, and evolutionary relationships. Definitions appear in Table 14.

*Table 14.* Concept definitions used for neuron analysis.

| Concept / Concept Group | Description |
| --- | --- |
| Repeat Tokens | We define a repeat-based concept. A token belongs to this concept if it appears within a repeated segment in one of the repeats present in the dataset. |
| Aligned Repeat Token to Mask Position | We define an alignment-based concept. A token belongs to this concept if it is aligned to the masked position, i.e., the token from which the model is expected to retrieve information to predict the masked token. This concept is motivated by manual inspection, which revealed consistent neuron patterns targeting this aligned position across multiple proteins. |
| Amino Acid Identity | We define amino-acid–based concepts by assigning a distinct concept to each of the 20 standard amino acids. A token is assigned to a concept if it represents that specific amino acid. This corresponds to a total of 20 concepts. |
| BLOSUM62-Based Concepts | We construct groups of substitutable amino acids using the BLOSUM62 substitution matrix (Henikoff & Henikoff, 1992). We view the matrix as a weighted graph over amino acids and identify cliques in which all pairwise substitution scores are greater than or equal to zero, corresponding to substitutions that are neutral or favorable. Each such clique is treated as a distinct concept. This procedure yields a total of 110 BLOSUM-based concepts. Although many of these cliques overlap, this does not pose an issue for our analysis, since each neuron is assigned to the best-matching concept according to its AUROC score. |
| Polarity (IMGT) | We define two polarity-based concepts according to the IMGT classification: *polar* and *non-polar*. A token is assigned to one of these concepts based on the polarity of the amino acid it represents. |
| Hydropathy (IMGT) | We define three hydropathy-based concepts according to the IMGT classification: *hydrophobic*, *neutral*, and *hydrophilic*. A token is assigned to one of these concepts based on the hydropathy class of the amino acid it represents. |
| Volume (IMGT) | We define five volume-based concepts according to the IMGT classification: *very small*, *small*, *medium*, *large*, and *very large*. A token is assigned to one of these concepts based on the volume class of the amino acid it represents. The *very small* (A, G, S) and *very large* (F, W, Y) classes also correspond to BLOSUM62-based substitution groups. Accordingly, neurons matched to these classes are categorized under the *BLOSUM* concept category in our analysis, even though they also reflect volume-based classification. |
| Chemical (IMGT) | We define seven chemically motivated concepts according to the IMGT classification: *aliphatic*, *aromatic*, *sulfur-containing*, *hydroxyl-containing*, *basic*, *acidic*, and *amide*. A token is assigned to one of these concepts based on the chemical class of the amino acid it represents. The *aromatic* (F, W, Y), *hydroxyl-containing* (S, T), *acidic* (D, E), and *amide* (N, Q) classes also correspond to BLOSUM62-based substitution groups. Accordingly, neurons matched to these classes are categorized under the *BLOSUM* concept category in our analysis, even though they also reflect chemical-based classification. |
| Physicochemical (IMGT) | IMGT defines eleven physicochemical classes based on combinations of hydropathy, volume, chemical properties, charge, hydrogen donor/acceptor capacity, and polarity of amino-acid side chains. Because most of these classes already covered by the Chemical and Amino-Acid categories, we retain only the aliphatic class, which is the only one introducing a concept not already captured by the previous categories. Note that the aliphatic class in the physicochemical IMGT scheme is more restrictive than the corresponding definition in the Chemical category. |

| Concept / Concept Group | Description |
|---|---|
| Aromatic Ring | We define an aromatic-ring–based concept consisting of amino acids whose side chains contain an aromatic ring. A token is assigned to this concept if it represents one of H, F, W, or Y amino acids. |
| Charge (IMGT) | We define three concepts derived from the IMGT classification: *positively charged*, *negatively charged*, and *polar uncharged*. For the latter, we do not use the full IMGT *uncharged* class ($A, N, C, Q, G, I, L, M, F, P, S, T, W, Y, V$), since this broad grouping produced many false negatives under manual inspection. Instead, we define *polar uncharged* as the intersection of the IMGT *uncharged* and *polar* classes, namely $S, T, N, Q, Y$. The *negatively charged* class also corresponds to a BLOSUM62-based substitution group, so neurons matched to this concept are categorized under the *BLOSUM* concept category in our analysis, even though the concept is also charge-based. |
| Hydrogen Donor / Acceptor (IMGT) | We define four concepts according to the IMGT classification based on hydrogen donor and/or acceptor properties of amino-acid side chains: *donor*, *acceptor*, *donor and acceptor*, and *none*. A token is assigned to one of these concepts based on the hydrogen bonding properties of the amino acid it represents. The *acceptor* class (D, E) overlaps with a BLOSUM62-based substitution group; accordingly, neurons matched to this class are categorized under the *BLOSUM* concept category in our analysis, even though they also reflect hydrogen-bonding properties. |
| Secondary Structure Propensity | We define four propensity-based concepts—*alpha helix former*, *beta sheet former*, *turn former*, and *helix breaker*—based on established secondary-structure tendencies (Chou & Fasman, 1978). A token is assigned to one of these concepts according to the amino acid it represents. The *alpha helix* formers (A, E, H, K, L, M, Q, R) preferentially stabilize helical structures, while the *beta sheet* formers (C, F, I, T, V, W, Y) preferentially stabilize beta-strand conformations. The *turn formers* (D, N, S) favor turn and loop conformations. Although Proline (P) and Glycine (G) also frequently occur in turns, we assign them to a distinct *helix breaker* class to account for their exceptionally strong destabilizing effects on helical structures (O'Neil & DeGrado, 1990). This separation was further motivated by manual inspection of important neurons in ESM-3, which frequently exhibited selective activation on Proline and Glycine residues. The *turn former* class (D, N, S) overlaps with a BLOSUM62-based substitution group; accordingly, neurons matched to this class are categorized under the *BLOSUM* concept category in our analysis. |
| Special Tokens | We define dedicated concepts for special tokens used by the model, including the beginning-of-sequence (BOS), end-of-sequence (EOS), and mask tokens. In addition, we define a joint concept that groups BOS and EOS to capture neurons that respond similarly to sequence-boundary tokens. A token is assigned to a special-token concept if it corresponds to one of these symbols, allowing us to distinguish structural and control tokens from amino-acid tokens in the neuron analysis. This design is motivated by manual inspection, which revealed neurons exhibiting similar activation patterns for these tokens. |

## J.3 Neuron Classification Details

To associate each neuron with its best-matching concept, we first collect its activations across all tokens in the *Approximate Repeats* dataset used in the circuit-discovery experiments, which contains approximately 3000 repeats. We exclude a small subset of sequences (16 sequences) in which more than one repeat segment type is eligible for analysis under our filtering criteria to simplify code for repeats concepts.

For a given concept, we partition tokens into a *concept group* and a *non-concept group*, and treat the matching task as a binary discrimination problem. We quantify the extent to which a neuron selectively responds to a concept by computing the

Area Under the Receiver Operating Characteristic curve (AUROC). Specifically, we scan over all distinct neuron activation values as decision thresholds and compute the corresponding true-positive and false-positive rates.

Prior work (Geva et al., 2021; Nikankin et al., 2025b; Bills et al., 2023) commonly evaluates neurons based on the tokens on which they are most strongly activated. Following this perspective, and accounting for the fact that neuron activations may be either positive or negative due to the gated linear unit structure of the MLPs, we define a signed AUROC-based score. For each neuron–concept pair, we first determine whether activations on tokens belonging to the concept group are predominantly positive or predominantly negative, based on the majority sign of the activations. If activations are predominantly positive, we use the AUROC directly as the neuron–concept score; if activations are predominantly negative, we use $1 - \text{AUROC}$. This convention allows us to capture whether a concept consistently drives the neuron toward either extreme of its activation range. For simplicity, we refer to this signed score as the AUROC throughout.

Each neuron is ultimately assigned to the concept with which it achieves the highest AUROC score. In some cases, concept groupings derived from IMGT and BLOSUM overlap. When such overlap occurs, we assign the neuron to the BLOSUM-based group for the purpose of reporting distribution-level AUROC plots, while noting that the neuron may reflect properties captured by both classification schemes.

**Exclusion of ambiguous tokens in concept evaluation.** To avoid ambiguity in concept assignment, we exclude certain tokens from the AUROC computation for specific concept classes. For repeat-related concepts (*Repeat Tokens* and *Aligned Repeat Source Token*), each sequence contains one repeat pair that satisfies our filtering criteria (Appendix B); however, additional repeat segments may exist in the same sequence and are excluded during dataset curation. Since the model may nevertheless recognize these additional repeats, all RADAR-identified repeat segments that were filtered out during dataset construction are omitted entirely from both the concept and non-concept groups. In addition, for each repeat segment—both the primary repeat under analysis and any excluded repeat segments—we exclude a window of $\pm 2$ tokens around the repeat boundaries identified by RADAR, as manual inspection revealed that neurons often activate on boundary-adjacent tokens as part of repeat-detection behavior.

For amino-acid– and biochemical–based concepts, we similarly exclude mask tokens from both the concept and non-concept groups.

**High-confidence thresholds for specific concepts.** For concepts related to *Special Tokens* and the *Aligned Repeat Token to Mask Position*, we impose a stricter matching criterion and assign a neuron to these concepts only if the corresponding AUROC score is at least $0.99$. We find that at lower thresholds, these concepts do not reliably capture the highly specific activation patterns expected of neurons selective for these concepts.

**Baseline.** In the standard procedure, for each concept we scan the entire dataset and assign all tokens that satisfy the concept definition to the concept set, with all remaining tokens forming the non-concept set. Neuron–concept alignment is quantified by computing the AUROC of neuron activations between these two sets.

As a baseline, we replace the true concept set with a randomly sampled set of tokens of equal size drawn from the same dataset. The AUROC is then computed in the same manner, comparing neuron activations on the randomly sampled tokens versus all other tokens. For each neuron, we record the highest AUROC obtained across concepts under this randomized setting.

## J.4    Additional Neuron Examples

Here we provide additional visualizations of neurons capturing different concepts, including both biochemical and repeat-related features.

**Aligned Repeat Token to Mask Position.** We visualize a neuron in ESM-3 located at layer 38, neuron index 2850, which achieves an AUROC of 0.993 on the *Aligned Repeat Token to Mask Position* concept. This neuron activates on the token aligned with the masked position—namely, the exact token the model is expected to retrieve as the correct answer. Interestingly, we did not identify any neuron exhibiting this behavior in the analyzed ESM-C model.

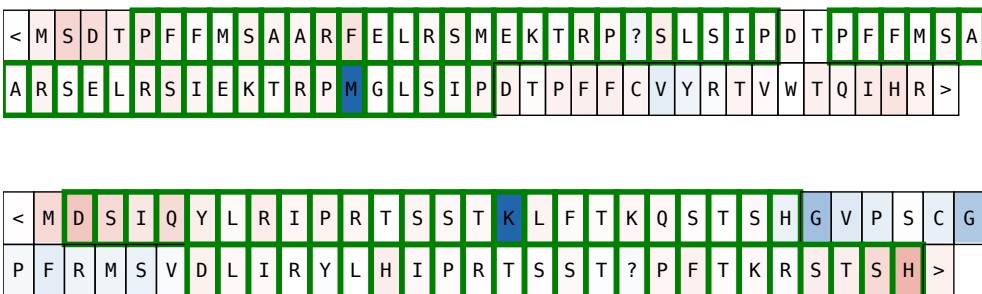

*Figure 25.* Example sequence visualizations for a neuron in ESM-3 (layer 38, neuron 2850) selective for the aligned repeat token to the masked position. The neuron activates on the token aligned with the mask (denoted by "?"), corresponding to the correct retrieval target. Repeat tokens are highlighted in green. UniProt accessions: A0A0P7XBW1 (top), B0DTM3 (bottom).

**Repeat Tokens.** We visualize a neuron in ESM-C located at layer 24, neuron index 2224, which achieves an AUROC of 0.92 on the *Repeat Tokens* concept. This neuron exhibits positive activations (red) on positions corresponding to approximate repeat tokens in the sequence, which are highlighted in green. This demonstrates that ESM-C, similarly to ESM-3, contains neurons that are selectively responsive to repeat structure.

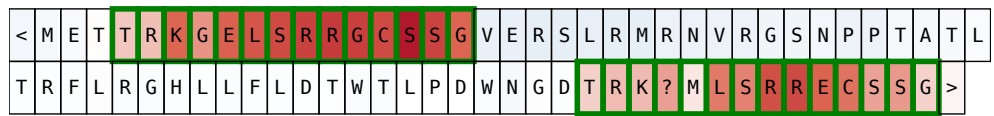

*Figure 26.* Example sequence visualization for a repeat-selective neuron in ESM-C (layer 24, neuron 2224). The neuron exhibits positive activations (red) at positions corresponding to approximate repeat tokens, highlighted in green. UniProt accession: A0A8X6HTE9.

**Helix breakers.** We visualize a neuron in ESM-3 located at layer 2, neuron index 3282, which achieves an AUROC of 0.997 on the *Helix Breakers* concept. This neuron exhibits positive activations (red) on the amino acids proline (P) and glycine (G), which are well known to disrupt $\alpha$-helical secondary structure due to their rigid (P) and highly flexible (G) backbones. We did not identify any neuron in ESM-C with AUROC greater than 0.75 for this concept.

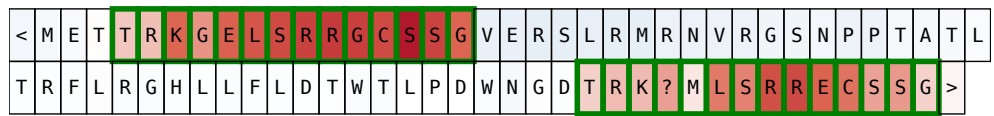

*Figure 27.* Example sequence visualization for a helix-breaker–selective neuron in ESM-3 (layer 2, neuron 3282). The neuron exhibits strong positive activations (red) on proline (P) and glycine (G), amino acids known to disrupt $\alpha$-helical secondary structure. UniProt accession: B0DTM3.

**Aromatic Ring.** We visualize a neuron in ESM-3 at layer 35, neuron index 3011, which achieves an AUROC of 0.963 on the *Aromatic Ring* concept. This neuron exhibits positive activations (red) on amino acids containing aromatic rings (F, Y, H, W).

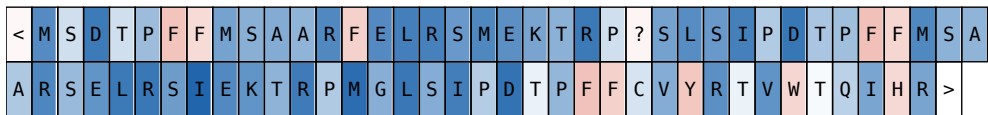

*Figure 28.* Example sequence visualization for an aromatic-ring–selective neuron in ESM-3 (layer 35, neuron 3011). The neuron exhibits strong positive activations (red) on aromatic amino acids (F, Y, H, W). UniProt accession: A0A0P7XBW1.

**Hydrogen Donor.** We visualize a neuron in ESM-C located at layer 7, neuron index 907, which achieves an AUROC of 0.995 on the *Hydrogen Donor (IMGT)* concept. This neuron exhibits positive activations (red) on amino acids arginine (R), lysine (K), and tryptophan (W), whose side chains can donate hydrogen atoms in hydrogen-bond interactions.

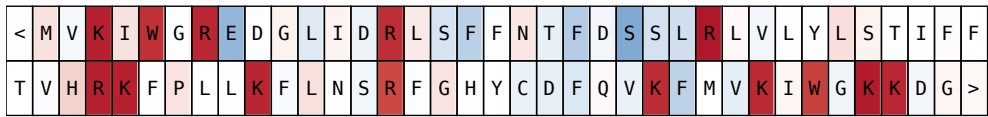

*Figure 29.* Example sequence visualization for a hydrogen-donor–selective neuron in ESM-C (layer 7, neuron 907). The neuron exhibits strong positive activations (red) on amino acids R, K, and W, consistent with hydrogen-donor side-chain chemistry. UniProt accession: A0A2M8A3Y9.

**Bidirectional Amino-Acid Neurons.** We observe many amino-acid neurons in both ESM-3 and ESM-C that exhibit strong activations with opposite signs for two different amino acids, indicating that amino-acid neurons can encode more complex features than simple single amino acid selectivity. Here, we visualize representative neurons from ESM-3 and ESM-C that demonstrate this bidirectional behavior.

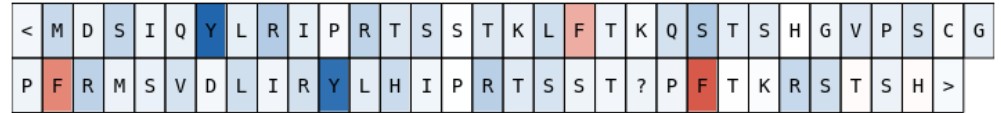

*(a)* ESM-3 (layer 8, neuron 788), exhibiting positive activations (red) on amino acid F and negative activations (blue) on amino acid Y. Notably, F and Y are biochemically similar aromatic residues with a high BLOSUM62 substitution score. UniProt accession: B0DTM3.

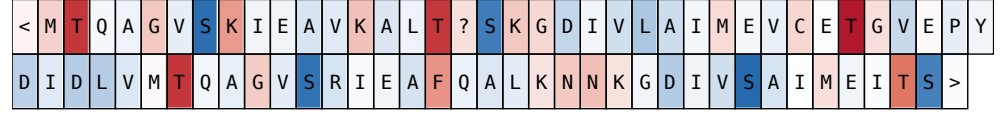

*(b)* ESM-C (layer 0, neuron 2105), exhibiting negative activations (blue) on amino acid S and positive activations (red) on amino acid T. Notably, S and T are often considered biochemically similar due to their hydroxyl side chains and have a high BLOSUM62 substitution score. UniProt accession: A0A2K3L8W6.

*Figure 30.* Example visualizations of bidirectional amino-acid neurons in ESM-3 (top) and ESM-C (bottom), showing strong but oppositely signed responses to pairs of amino acids.

**BLOSUM62.** We visualize additional neurons in ESM-3 and ESM-C associated with BLOSUM62 substitution cliques, where each clique represents a group of amino acids that are mutually substitutable according to the BLOSUM62 matrix. Specifically, we show a neuron in ESM-3 at layer 19, neuron index 3434 (AUROC = 0.963), and a neuron in ESM-C at layer 0, neuron index 2615 (AUROC = 0.988), both selective for the *BLOSUM62 clique* {D, E, N}.

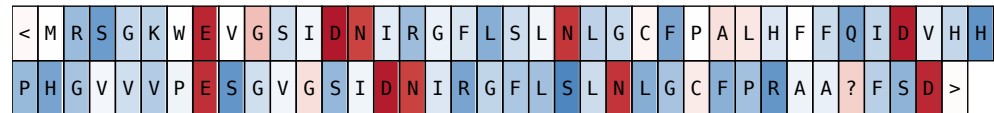

*(a)* ESM-3 (layer 19, neuron 3434), exhibiting positive activations (red) on amino acids D, E, and N. UniProt accession: A0A0P7XBW1.

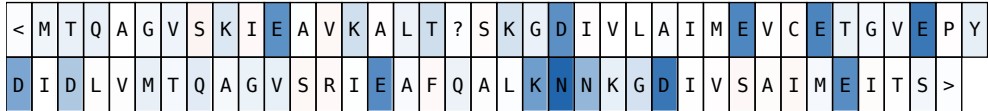

*(b)* ESM-C (layer 0, neuron 2615), exhibiting negative activations (blue) on amino acids D, E, and N. UniProt accession: A0A2K3L8W6.

*Figure 31.* Example sequence visualizations for neurons selective for the BLOSUM62 substitution clique {D, E, N} in ESM-3 (top) and ESM-C (bottom).

**Special Tokens.** We visualize neurons in ESM-3 and ESM-C associated with special tokens. Specifically, we show a neuron in ESM-3 at layer 0, neuron index 125 (AUROC = 1.0), and a neuron in ESM-C at layer 24, neuron index 2127 (AUROC = 0.99), both selective for the *BOS token* concept.

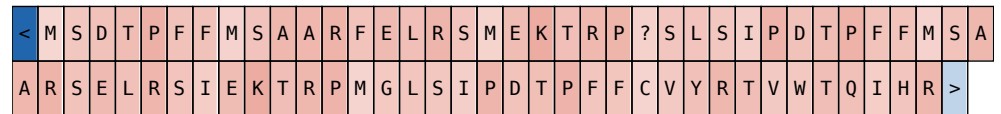

*(a)* ESM-3 (layer 0, neuron 125), exhibiting strong negative activations (blue) on the BOS token. Regular amino-acid tokens show positive activations (red), while the EOS token exhibits moderately negative activation. UniProt accession: A0A0P7XBW1.

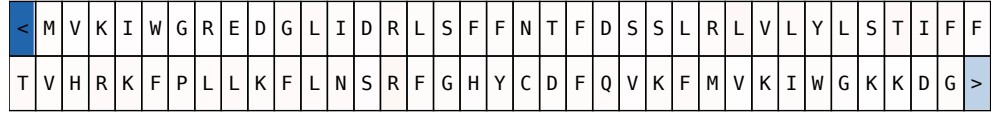

*(b)* ESM-C (layer 24, neuron 2127), selectively activating on the BOS token with strong negative activation (blue). The EOS token also exhibits moderately negative activation. UniProt accession: A0A2M8A3Y9.

*Figure 32.* Example sequence visualizations for neurons selective for special tokens in ESM-3 (top) and ESM-C (bottom). Both neurons exhibit strong, selective activations to the BOS token, with weaker activations to the EOS token. In the visualizations, < denotes the BOS token and > denotes the EOS token.

