# OpenReview forum: "Induction Meets Biology: Mechanisms of Repeat Detection in Protein Language Models"
_ICML.cc/2026/Conference — ICML 2026 regular_

### Official Review · Reviewer_z4Ji · 2026-02-21

**Soundness:** 4
**Presentation:** 4
**Significance:** 3
**Originality:** 4
**Overall Recommendation:** 6
**Confidence:** 5

**Summary:**

This work employs a variety of techniques, spanning bioinformatics annotation tools for repeat discovery and reverse engineering tools for neural networks, to investigate the mechanisms by which protein language models (PLMs) generate repetitive protein sequences of the kind observed in nature.

The reverse engineering tools employed include attribution patching with integrated gradients (AP-IG) to discover circuits by investigating the response of activations to counterfactual inputs, the related EAP-IG, Intersection-over-Union (IoU) to investigate circuit overlap, manual inspection and automated clustering of attention heads (followed by Elbow/inertia and Silhouette analyses), and a number of statistics computed for attention heads (induction score, relative position score, amino-acid bias, summarization of various biophysical amino acid properties, repeat focus, attention to beginning/end of sequence, attention entropy, context sensitivity, relative contribution to residual stream, vocabulary entropy).

The authors find that PLMs model repeats by similar mechanisms to those used by LLMs to copy substrings, with additional biologically specialized components to model biophysically relevant features and evolutionarily common substitutions.

**Compliance With Llm Reviewing Policy:**

Affirmed.

**Final Justification:**

Rebuttal increased appreciation for work.

**Key Questions For Authors:**

I am curious to see how the results break down by repeats of varying length and complexity.

Also, obviously, very curious about >2 repeats, but that seems to be out of scope of the work as presented.

**Limitations:**

yes

**Strengths And Weaknesses:**

Investigation of copying (and related tasks) has been an important tool in the LLM reverse engineers’ toolkit. This work essentially introduces the same idea to understanding PLMs. As such I think it is quite significant.

The reverse engineering tools employed are quite interesting, and for me, will be a model reference for how to approach an understanding of the internal workings of PLMs. I personally find inspection of network internals to be a useful debugging tool, and reasoning about those internals helps my understanding. I am always pleased to see work like this, that moves beyond the idea of networks as a magical opaque component.

Further, this work strengthens the connections between PLMs and LLMs.

With that said, this is (necessarily) an empirical investigation, and really (by the authors’ own admission in the Conclusions & Limitations) a “first step” rather than a comprehensive treatment. The authors have only examined fairly simple situations: exactly two repeats, <50% substitutions. Real proteins contain repeats that are often multi-copy. The authors acknowledge these limitations.

The authors cite several biologist-oriented reviews of repeats, as well as some tools (RADAR, HHrepID). This is to be commended, because in an empirical investigation like this, it adds great value to make use of the existing domain-specific knowledge. There is a rich literature on bioinformatics tools and databases for annotating repeats (a few more examples: Repetita, SOLeNNoiD, STRPsearch), and it may be helpful for the authors to dig deeper into this literature as they continue this research.

On balance, I find this work important and useful. It is good for machine learning research to spend time trying to understand what its black boxes are doing, as opposed to continually trying to build bigger, better, and (typically) even harder-to-understand machines.

On the other hand, the work is a first step rather than a comprehensive treatment, and is up against some stiff competition. Interesting and novel as the reverse engineering is, it would benefit from going a bit deeper; in particular, investigating how the copying mechanism is modified when there are more than two copies of a repeat. There are many unanswered questions that remain, e.g. does the network focus on one template, or does it diffuse attention across all the copies? It is also interesting to ask about the information complexity of repeats generated by PLMs compared to real sequences, and how this extends to multi-copy repeats. Finally one might spend more time distinguishing between repeats of various lengths: microsatellites, coiled-coils, beta-barrels, and repeated domains are quite different entities.

Due to this being a (fascinating) first step into a (much bigger) area, I think this work would benefit from a little more maturity before publication in a top ML conference. Nevertheless, I rate this highly as a solid and interesting empirical study.

---

> ### Author Rebuttal · Authors · 2026-03-27
>
> Thank you for your review!
>
> We appreciate that you find our work significant, useful, important for the community, and view it as a valuable tool for understanding PLMs – this was a strong motivation for this line of research.
>
> We also thank the reviewer for mentioning additional bioinformatics tools for further use (additional to the ones we relied on in our research).
>
> To address the reviewer’s listed questions:
>
> - **”...how do the results break down by repeats of varying length and complexity”**
>
> We analyzed performance across repeat length and substitution rate (up to 50%), and observed a clear trend: both models perform worse on shorter repeats with higher substitution rates, and improve substantially as repeat length increases (see our response to Reviewer Jn46 for full results).
> In addition, a stress-test experiment in which we gradually introduced substitutions into identical repeats, thereby converting the setting into an approximate-repeat one, shows that the induction mechanism becomes less important for the repeat token prediction task at lower substitution rates in shorter repeats (e.g., ~30% substitutions for length 10 vs. ~70% for length 30), providing a possible explanation for the lower performance observed on shorter repeats at higher substitution rates.
>
> - **”how is the copying mechanism modified when there are more than two copies of a repeat? does the network focus on one template, or does it diffuse attention across all the copies?”**
>
> This is a set of great questions! We definitely think about these as directions for further work. As the reviewer mentions, while our current study focuses on the simpler, 2-repeat setting, it is (as far as we know) the first mechanistic study on repeat mechanisms in PLMs. We believe understanding this basic setting predates more complex questions and is required as a good basis to answer further questions on more complex settings.
>
> We hope our answers are satisfactory. Please let us know if you have any other questions or concerns.

---

> > ### Author Rebuttal · Reviewer_z4Ji · 2026-03-31
> >
> > These are interesting answers thank you.

---

> > > ### Author Response · Authors · 2026-04-03
> > >
> > > We thank the reviewer again for the thoughtful review and for indicating that our responses addressed the questions raised. We appreciate the positive assessment of our work and the helpful suggestions for future directions.

---

### Official Review · Reviewer_pHHB · 2026-02-22

**Soundness:** 2
**Presentation:** 3
**Significance:** 2
**Originality:** 3
**Overall Recommendation:** 4
**Confidence:** 4

**Summary:**

This paper aims to elucidate the internal mechanisms of PLMs, focusing on ESM-3 and ESM-C, analyzing the case study of repeats. To do this, the authors aim to discover circuits within PLMs that govern these predictions, testing on synthetic data, real-world identical repeats, and real-world approximate repeats. Through their experiments, the authors find that PLMs learn basic biochemical and contextual representations in early layers, repeating contexts in the middle layers, and masked-prediction specific representations in the later layers.

**Compliance With Llm Reviewing Policy:**

Affirmed.

**Final Justification:**

Upon review of the author's rebuttal, I had missed the part of the paper where the attribution scores were aggregated across samples, as well as on different masked tokens. This alleviates my concerns given in W2 and makes the analysis stronger. I believe the authors have adequately addressed my comments, and while I still remain skeptical about the broad scope of the paper, I believe the analysis on this specific case study is enough for me to raise my score.

**Key Questions For Authors:**

1. Would this method work for other applications of PLMs, or is it mainly tailored for repeats?
2. What might a mechanistic understanding of repeats enable? Is it mainly to further our biological understanding, or are there relevant applications which might require this understanding?
3. Do you find that the discovered circuits in PLMs remain fixed no matter which amino acid one masks?

**Limitations:**

The limitations section in the paper is thoughtfully written, except for some points that I have raised above.

**Strengths And Weaknesses:**

Strengths
- The paper is clear and well-written. The paper appears to be technically sound.
- Circuit discovery in PLMs is a relatively novel and underexplored area.
- The authors provide a systematic analysis of a biologically-relevant case study. Motivation across the paper and experiments is clear, and written in sufficient detail to reproduce.
- Biologically plausible explanations are presented thoughtfully and backed up with sufficient evidence.

Weaknesses
- My main concern with this paper is its limited scope. The paper primarily focuses on understanding PLMs on repeats, but it is unclear how generalizable these methods are to realistic usage cases of PLMs (e.g., function prediction). It is also unclear whether the internal mechanisms/general PLM logic uncovered will also carry over to more complex tasks.
- In the paper, it is written that the authors “mask a single amino acid token” and aim to discover the circuit for the PLM that predicts the masked token. However, the choice of discovering a circuit given a single token feels arbitrary. As I understand it, given two repeating segments each of length $n$ (like in Figure 1), there are theoretically $n$ possible ways to discover a circuit in a single segment, and $2 \times n$ possible ways to discover a circuit if you consider the other segment.
- Adding onto the above point, it’s not clear whether the prediction of a single token is truly representative of a PLM’s logic governing repeats. For example, if one were to discover a circuit by masking a known binding site, where a specific amino acid is known to facilitate the binding of a ligand/protein, then it would be fair to say that the circuit that predicts the correct amino acid likely is representative of a PLM’s understanding of binding. However, there is no single amino acid that serves as a basis to discover a circuit in repeats.

Overall, the paper appears to be technically sound and the methodology is novel, but I have concerns about the paper’s biological motivations and scope, which is the central focus of the paper. I have left several questions below, and I would appreciate it if the authors took time to address my concerns.

---

> ### Author Rebuttal · Authors · 2026-03-27
>
> Thank you for your review!
> We appreciate that you find our work to be novel, technically sound, and our analyses biologically convincing.
>
> Regarding the listed weaknesses and open questions:
>
> - **Generalizability of the method to other realistic PLM use cases (e.g., function prediction) / generalizability of the mechanism to more complex tasks**
>
> First, we would like to emphasize again the importance of repeats in biology, as discussed in Appendix A. Many repeat types are directly linked to protein function, and repeats also serve as an important evolutionary mechanism. We therefore view this as a biologically realistic setting, and believe the mechanism we identify may form part of broader function-related behavior.
>
> Second, induction mechanisms in language models have been linked to a range of complex in-context behaviors [1], suggesting that such mechanisms may also support broader sequence-processing in PLMs.
>
> Third, the method is not specific to repeats. It builds on standard mechanistic analysis techniques from LMs [2,3] and can, in principle, be applied to any PLM task for which a suitable performance signal and counterfactual can be defined. In pretrained PLMs, one natural way to do this is through masked token prediction, while in fine-tuned models the same framework can be applied directly to the downstream prediction.
>
> - **The choice of discovering a circuit given a single token feels arbitrary/ it’s not clear whether the prediction of a single token is truly representative of a PLM’s logic governing repeats/ Are the discovered circuits remain fixed?**
>
> To discover a circuit for each task, we ran the model on 500 samples that varied in sequence, masked position, and repeat length. The masked position was randomly selected from the repeat tokens, and attribution scores were aggregated across samples, yielding a single circuit shared across examples. Circuit size was then selected based on 85% faithfulness on a separate evaluation set of 500 samples. Thus, the discovered circuit is not tied to one token or one repeat, but generalizes across positions and repeat instances.
>
> Regarding whether single-token prediction is representative of repeat detection, Appendix F.2 provides supporting evidence that it is. Corrupting one of the repeated segments substantially harms prediction of the masked token, whereas corrupting a matched non-repeat segment has almost no effect. This indicates that correct prediction depends on recognizing the repeat pattern rather than only on local cues.
>
> - **What might a mechanistic understanding of repeats enable? Is it mainly to further our biological understanding, or are there relevant applications which might require this understanding?**
>
> The main goal is indeed to improve biological understanding. More specifically, this mechanistic analysis helps us understand what PLMs learn about repeat detection and whether these internal mechanisms align with what we know about biology. In our case, we find meaningful biological structure, for example through the emergence of biochemical-similarity neurons.
>
> This analysis may also support practical applications. One direction is improved repeat detection: recent work has used PLM embeddings for repeat detection [4], and our findings suggest that repeat-related components, such as induction heads or repeat neurons, may help improve such pipelines.
>
> A second application is better understanding and mitigation of model behavior in repeat-related settings. For example, recent work has shown differences in structure prediction across isoforms, which are highly similar versions of the same protein [5]. While our setting studies repeats within a single sequence, a similar mechanistic approach could be extended to “repeats across sequences,” for example by using one isoform as a counterfactual for another, in order to better understand what drives these prediction differences.
>
> Finally, our analysis is also relevant to repetition effects in PLMs more broadly. Recent work suggests that repetition can harm the information content of protein embeddings for downstream tasks [6], and another work studies pathological repetition in PLM generation and proposes mitigation via steering [7]. Our findings provide a mechanistic view of the repeat-related components that may underlie these phenomenas, and could help motivate targeted interventions, similar to mitigation ideas explored in language models [8].
>
> We hope these clarifications address the reviewer’s concerns and better motivate the biological relevance, applicability, and broader significance of our analysis.
>
> [1] https://arxiv.org/abs/2209.11895
>
> [2] https://arxiv.org/abs/2304.14997
>
> [3] https://arxiv.org/abs/2410.21272
>
> [4] https://www.biorxiv.org/content/10.1101/2024.06.07.596093v1
>
> [5] https://www.pnas.org/doi/10.1073/pnas.2406285121
>
> [6] https://arxiv.org/pdf/2504.17068
>
> [7] https://arxiv.org/abs/2602.00782
>
> [8] https://arxiv.org/abs/2507.07810

---

> > ### Author Rebuttal · Reviewer_pHHB · 2026-04-01
> >
> > Thank you for the comments and clarifications. It is clear I missed the part of the paper where the attribution scores were aggregated across samples, as well as on different masked tokens. This makes the analysis more convincing for me, and I am raising my score to a 4. I wish the authors the best of luck!

---

> > > ### Author Response · Authors · 2026-04-03
> > >
> > > We thank the reviewer again for the thoughtful review and for acknowledging that our responses addressed the questions raised and clarified the main concerns.

---

### Official Review · Reviewer_mGgi · 2026-03-02

**Soundness:** 3
**Presentation:** 2
**Significance:** 3
**Originality:** 4
**Overall Recommendation:** 5
**Confidence:** 4

**Summary:**

The authors conduct a mechanistic interpretability study in order to understand how protein language models detect repeat/near-repeat patterns in protein sequences. The paper employs a series of observational and causal mech interp techniques and finds  computational structures 1) that emerge in causal language models (e.g. induction heads)  and 2) that carry biologically relevant information (e.g. neurons coding for AA similarity). Induction heads and MLPs appear to be the largest drivers of repeat detecting behaviour.

**Compliance With Llm Reviewing Policy:**

Affirmed.

**Final Justification:**

As I have pointed out in my original review, I believe this is a solid paper with interesting findings for multiple communities within the ML space. I chose to keep my score at a 5 and not raise it to a 6 because I think the paper could have been written more clearly

**Key Questions For Authors:**

- Are there relative position heads for different lengths? Do they always come before the biochemical similarity neurons?
- Why are the AA specific neurons necessary after induction heads? Can the masked AA not be inferred from induction alone?
- Are insertion and deletion mutations common in biology? If so, what do you make of the models not being able to predict these?
- Why do you think the repeat tokens have activities in such high magnitude? It seems like in Figure 6, sometimes they go as low as -50.
- Why does the synthetic circuit generalise better to the identical circuit and not the other way around? Is it because the synthetic cannot rely at all on pre-existing knowledge?
- What does AA based attention mean? is it structured or a random mapping between fixed source and destination tokens?

**Points about presentation**
- Figure 1 is very confusing. Among other things, I don't understand why some arrows are misaligned (the green and the blue).
- There is an excessive use of em-dashes, which at times interrupts the flow of the text.
- It's not clear whether some analyses presented in the main text are complementary or redundant. For example, while it's good to show cross-task circuit comparisons using several methods, I don't think having to work through three different methods in the main text helps to convey the message of the paper. Some can go to the appendix. Similarly, does Figure 8 tell the reader something new that we cannot infer from Figure 7?
- The changing styles in plots (inconsistent colour mappings, quality, the underlying grid) make the figures sometimes hard to follow.

**Limitations:**

- There is a bit of a mismatch between some of what's written in the main text and the appendix. For example, section 4.1 says the authors found $k=3$ to be optimal for clustering via elbow and silhouette analyses. However, the appendix shows that the silhouette score does not draw such a clear picture.
- Being interested in mechanistic interpretability, I found the findings exciting. However, it is unclear whether these findings will enhance our understanding of biology or downstream use of these models.

**Strengths And Weaknesses:**

## Strengths
- The empirical findings are interesting to several communities (bioinformatics, and mech interp).
- There are surprising similarities to causal language models, such as induction heads. To my knowledge, this has not yet been shown in non-causal language models.
- Authors support their claims through the use of several different mech interp methods, both causal and observational

## Weaknesses
- I find the presentation of the paper less than ideal. Even though it's great that authors use several methods for analysis, it is not always clear what the point of each analysis is in the main text. I provide more detailed feedback on this in the questions section below.  I believe addressing these points requires a major rewrite, which I don't think would fit in the timeline of rebuttals.  Therefore, it is highly unlikely I will raise my score.

---

> ### Author Rebuttal · Authors · 2026-03-27
>
> Thank you for the review!
> We appreciate you found our findings interesting, novel and well-supported.
>
> To answer your questions:
> - **Are there relative position heads for different lengths? Do they always come before the bio-similarity neurons?**
>
> We observe multiple relative position heads across layers, attending to both backward and forward offsets of varying lengths. They do not consistently appear before biochemical similarity neurons; instead, both are distributed across the model. However, the early layers are dominated mainly by these two component types.
>
> - **Why are the AA specific neurons necessary after induction heads?**
>
> These neurons may adjust the final probability of the correct amino acid by increasing or decreasing its logit after induction heads promote it. Similar roles for late-layer neurons have also been observed in language models [1].
>
> - **Are insertion and deletion mutations common in biology? If so, what do you make of the models not being able to predict these?**
>
> Insertions and deletions are generally much less common than substitutions [3]. In our qualitative inspection, indels often cause induction heads to misalign the repeated segments, leading attention to drift and preventing the masked token from aligning with the correct residue.
>
> - **Why do you think the repeat tokens have high magnitude activations?**
>
> It is not meaningful to directly compare activation magnitudes across different neurons, but only across tokens for the same neuron. Under our key–value decomposition, each neuron contributes activation × output vector to the token representation, and output-vector norms vary across neurons (Appendix I). Thus, larger activations do not necessarily indicate stronger influence.
>
> - **Why does the synthetic circuit generalise better to the identical circuit and not the other way around? Is it because the synthetic cannot rely at all on pre-existing knowledge?**
>
> Yes, one possible reason is that the synthetic setting is harder because the sequences are out of distribution. In addition, our circuit discovery relies on counterfactual replacements with similar amino acids; these replacements may remain biologically meaningful in natural sequences, but be less meaningful in synthetic ones, making the intervention noisier and potentially requiring more components for recovery.
>
> - **What does AA biased attention mean?**
>
> We observe structured patterns rather than random mappings. These heads tend to assign higher attention to specific subsets of amino acids as key tokens, without correlation to the fixed positions in the sequence. In early layers, some heads attend broadly from many query tokens to all instances of certain amino acids, while in mid-to-late layers the patterns become more specific, often mapping particular queries (e.g., repeat tokens) to specific amino acid keys. We provide examples in Appendix H.
>
> - **the appendix shows that the silhouette score does not draw such a clear picture.**
>
> For ESM-3, both the elbow method and the silhouette score support $k=3$. For ESM-C, the elbow method supports  $k=3$, while the silhouette score slightly favors $k=2$. Given this ambiguity, we also qualitatively inspected the resulting ESM-C clusters. We found that  $k=3$ yields a clearer functional separation, in particular by distinguishing amino-acid-biased heads from relative-position heads, which are merged when $k=2$. We further verified in Appendix H.4.2, for both models, that the clusters remain distinct using a statistical test.
> We will clarify this point in the main paper in the next revision.
>
> Regarding your comments on presentation:
> - **"...I don't understand why some arrows are misaligned (green and blue)"**
>
> 	In Fig 1, the green arrows indicate that stage 2 components (marked in green) generally occur in earlier layers then stage 3 components (marked in blue). We thank the reviewer for pointing this out, and will restructure the figure to convey our point better.
>
> - **The changing plot styles and excessive use of em-dashes affect readability.**
>
> Thank you for noting this. We will standardize plot colors across identical groups and reduce the use of em-dashes in the revised version.
>
> - **Regarding the three cross-task comparison metrics**:
> We agree that IoU and recall are complementary. However, cross-task faithfulness measures functional rather than structural similarity, so it may lead to different conclusions [2]. This distinction was important for our goal in this section.
>
> - **Regarding Figures 7 and 8**:
> Figure 7 shows interaction strengths between component groups, whereas Figure 8 shows where these groups promote the correct answer across layers and which group acts first.We will clarify these two points in the revised text.
>
> We hope our answers are satisfactory. Please let us know if you have any other questions or concerns.
>
> References:
>
> [1] https://arxiv.org/pdf/2406.19384
>
> [2] https://arxiv.org/abs/2506.09047
>
> [3] https://pubmed.ncbi.nlm.nih.gov/19329651

---

> > ### Author Rebuttal · Reviewer_mGgi · 2026-03-31
> >
> > As I have written below, the authors have thoroughly answered my questions. I maintain my accept score (5) and do not raise because:
> >
> > As I have written in my original review, the paper could be better structured in the main text with an easier to follow narrative and more lean analyses

---

> > > ### Author Response · Authors · 2026-04-03
> > >
> > > We thank the reviewer again for the thoughtful review and for acknowledging that our responses addressed the questions raised.  We appreciate the positive evaluation and the constructive feedback regarding the presentation and structure of the paper, and we will take this into account in the revised version.

---

### Official Review · Reviewer_Jn46 · 2026-03-09

**Soundness:** 2
**Presentation:** 3
**Significance:** 3
**Originality:** 3
**Overall Recommendation:** 4
**Confidence:** 3

**Summary:**

This paper studies how protein language models (PLMs) detect and exploit repeated sequence segments (“repeats”) for masked-token prediction. The authors define three repeat settings—synthetic exact repeats, natural identical repeats, and natural approximate repeats—and primarily evaluate on ESM-3, with additional evidence on ESM-C. They apply an attribution patching based circuit discovery procedure (Attribution Patching with Integrated Gradients, AP-IG) to identify attention heads and MLP neurons causally important for repeat-based recovery. The paper argues that 1) repeat processing relies on a staged mechanism combining alignment/position-sensitive components, induction like attention that copies information across repeated segments, and neurons sensitive to amino-acid similarity; and 2) approximate repeat circuits functionally subsume exact/identical-repeat circuits via cross-task faithfulness.

**Compliance With Llm Reviewing Policy:**

Affirmed.

**Final Justification:**

The research problem is interesting, and the rebuttal has addressed most of my concerns; however, I think more experiments across a broader family of PLMs are necessary to assess the generalizability of the proposed method.

**Key Questions For Authors:**

Here are some key questions:

1. **Robustness to counterfactual choice:** How stable are the discovered circuits (key heads/neurons, faithfulness) under alternative counterfactuals not derived from BLOSUM62?
2. **Failure modes:** For approximate repeats, what are the dominant error categories? Do attention patterns differ qualitatively between successes and failures (e.g., alignment drift, distractor attention)?
3. **Generality across PLMs:** Do you see similar induction-like heads and similarity neurons in non-ESM PLMs (different architectures/objectives/tokenization)? You may need to add more protein language model families and architectures(autoregressive models like: protgpt2[1], progen[2], progen2[3]; diffusion lm: dplm[4], dplm2[5])


[1]Ferruz, N., Schmidt, S. & Höcker, B. ProtGPT2 is a deep unsupervised language model for protein design. Nat Commun 13, 4348 (2022). https://doi.org/10.1038/s41467-022-32007-7
[2]Madani, A., Krause, B., Greene, E.R. et al. Large language models generate functional protein sequences across diverse families. Nat Biotechnol 41, 1099–1106 (2023). https://doi.org/10.1038/s41587-022-01618-2
[3]ProGen2: Exploring the boundaries of protein language models
Nijkamp, Erik et al.
Cell Systems, Volume 14, Issue 11, 968 - 978.e3
[4]Wang, Xinyou et al. “Diffusion Language Models Are Versatile Protein Learners.” International Conference on Machine Learning (2024).
[5]Wang, Xinyou et al. “DPLM-2: A Multimodal Diffusion Protein Language Model.” ArXiv abs/2410.13782 (2024): n. pag.

**Limitations:**

Please see questions; the authors can improve the paper following the suggestions above in "Key Questions"

**Strengths And Weaknesses:**

Strongness:
1. **Important and well-motivated problem.** Repeats are widespread in proteins and relevant to structure/function/evolution; understanding PLM behavior on repeats is valuable for trust and downstream use.
2. **Task suite spans controlled and realistic settings.** The combination of synthetic and natural repeats is a solid design for mechanistic analysis while remaining biologically grounded.
3. **Relatively standard circuit discovery pipeline.** Using AP-IG with faithfulness-based evaluation is stronger than purely correlational probing, and the “minimal faithful circuit” framing is helpful.
4. **Clear mechanistic hypothesis linking to induction heads.** Drawing a parallel to induction in NLP and connecting it with amino-acid similarity representations is conceptually appealing.

Weakness:
1. **Counterfactual construction may bias the discovered circuits.** The approach relies heavily on BLOSUM62-driven substitutions to create counterfactuals(line128). While biologically reasonable, this may “inject” an explicit similarity prior into the intervention and preferentially surface BLOSUM-aligned features/neurons. Since you are finding a circuit using the attributing method(here in your paper is AP-IG), you need to make sure the counterfactual pairs are only different in what you want to attribute.  The paper would benefit from a more systematic robustness study across counterfactual types (e.g., random substitutions) and from reporting circuit stability.
2. **Causal validation could be strengthened.**  AP-IG is an improvement over probes, but direct interventions would better support the mechanistic narrative.
3. **Approximate-repeat performance is substantially lower, but failure analysis is limited.** In Table 1, about 79% accuracy implies a relatively large fraction of failures. The mechanistic story is primarily built on successful instances; it remains unclear what the failure mode of the approximate repeat task is. In my opinion, the approximate repeat prediction task is not well designed. See line 101("We restrict the substitution rate to at most 50%"). I am not sure whether this will lead to a distribution misalignment.

Concerns:
1. **Distinguishing “protein specific” from “generic sequence induction” remains ambiguous.** The induction copying story may generalize to any symbolic repeat task. The paper argues for biology specific components (amino acid similarity neurons), but it is not fully convincing that these are uniquely protein-related rather than generic learned token similarity.

---

> ### Author Rebuttal · Authors · 2026-03-27
>
> Thank you for the review!
> We appreciate that you found our work timely, well-motivated and well grounded.
>
> To address your questions and weaknesses:
>
> - **Robustness to counterfactual choice**
>
> Counterfactuals are indeed a major source of bias in circuit analysis. In Appendix F, we compare several counterfactual choices and find that they recover a shared mechanism with high structural overlap, while BLOSUM-based counterfactuals yield more compact circuits. These comparisons were originally conducted only at the attention-head and MLP level. After the submission deadline, we ran a neuron-level analysis using the permutation counterfactual from Appendix F. We found that 83.29% of the neurons previously discovered with BLOSUM counterfactuals were also recovered with permutation counterfactuals. Among neurons previously matched to BLOSUM62 and other biological concepts, 73.5% were also rediscovered, suggesting that these neurons are not specific to a single counterfactual choice.
>
> - **Direct interventions would better support the mechanistic narrative (more than AP-IG)**
>
> We agree that direct interventions can sometimes provide more faithful evidence. However, attribution patching with integrated gradients (AP-IG) is a widely used approximation to causal analysis [1,2,3,4], and has been shown to often achieve comparable faithfulness to direct interventions on circuit-discovery benchmarks [5]. Nevertheless, as per standard procedure, we measure faithfulness to validate that the circuits we find indeed account for most (>85%) of the behavior we are localizing.
>
> - **Approximate-repeat setting failure analysis**
>
> We would like to point out that mechanism analysis relies on a base assumption of success. Thus, analyzing failure modes was not our main focus.
> However, we ran additional analyses to identify cases in which the models fail.
>
> First, we re-aggregated the original approximate-repeat performance by grouping sequences into bins defined by repeat length and substitution rate, and averaged the scores within each bin.
>
> **Results for ESM3 (ESM-C shows a similar trend across):**
>
> | Substitution % \ Length | 1–10 | 11–15 | 16–20 | 21–25 | 26+ |
> |------------------------|------|-------|-------|-------|-----|
> | 1–25%                  | 0.77 | 0.94  | 0.96  | 0.98  | 0.98 |
> | 25–50%                 | 0.45 | 0.63  | 0.85  | 0.92  | 0.96 |
>
> These results suggest that both models fail primarily on short repeats with high substitution rates, and perform substantially better as repeat length increases.
>
> Furthermore, we ran a stress-test experiment on natural proteins with identical repeats of lengths 10, 20, and 30. For each sample, the task was to predict a masked token within a repeat, as in the original identical-repeat setting. We gradually mutated a repeat using the most likely BLOSUM substitutions, and measured the induction heads’ average AP-IG score versus the number of mutations.
> The results show that induction heads stop being important for masked-token prediction within repeats at ~30% substitutions for length 10, 60% for length 20, and about 70% for length 30. These results provide a possible explanation for the weaker model performance on shorter repeats and are consistent with our evaluation results, indicating that the model relies on induction for repeat pattern identification.
>
> Following the reviewer’s request, we also qualitatively examined failed examples. In successful cases, repeat tokens attend strongly to their aligned counterparts. In failed cases, the head either attends to BOS/EOS tokens instead of the repeat, or only partially recognizes the repeat with weak attention between aligned positions.
>
> - **Generality across PLMs**
>
> We focused on encoder-based PLMs, specifically the widely used ESM family. Importantly, the two models we analyze already differ in architecture, training data, and training regime; ESM-3 uses very high masking ratios during training, similar to setups of diffusion-based PLMs. As shown in Figure 6C and discussed in Section 4, we did find differences between those two models. Extending the analyses to additional PLM families, especially autoregressive models, is an important direction for future work, but would require redefining the task and adapting the evaluation.
>
> - **50% substitution rate choice**.
>
> Our goal was to create a controlled and simplified setup for mechanistic analysis, following many prior works on language models. This choice is also biologically motivated, since highly diverged repeats are harder to detect from sequence alone and often require structural information [6].
>
> We thank the reviewer again, and we will add these results to the revised version. Please let us know if you have any further questions.
>
> [1] https://arxiv.org/abs/2502.00873
>
> [2] https://arxiv.org/abs/2506.09047
>
> [3] https://arxiv.org/abs/2407.10827
>
> [4] https://arxiv.org/abs/2510.11210
>
> [5] https://arxiv.org/abs/2504.13151
>
> [6] https://www.nature.com/articles/srep23959

---

> > ### Author Rebuttal · Reviewer_Jn46 · 2026-04-01
> >
> > You have mostly addressed my concerns. I will increase the score to 4. But I still want to more experiments/analysis in AR models like protgpt2, progen or DLM like DPLM. This can be extented in your future work. Tks.

---

> > > ### Author Response · Authors · 2026-04-03
> > >
> > > We thank the reviewer again for the thoughtful review and for acknowledging that our responses addressed the main concerns. We agree that extending the analysis to additional protein language model families is an important direction for future work.

---

### Decision · Program_Chairs · 2026-04-30

**Decision:**

Accept (regular)

**Comment:**

This study provides an interesting mechanistic understanding of how protein language models (ESM-3 and ESM-C) detect and exploit sequence repeats. The authors apply attribution patching with integrated gradients (AP-IG) to discover circuits responsible for repeat-based recovery, identifying a staged mechanism involving induction heads and neurons sensitive to amino acid similarity. The paper is technically solid, well-motivated, and valuable for both the biology and language model communities.

**Strengths:**

- Well-motivated problem with biological relevance. (Jn46, mGgi, z4Ji)
- Rigorous combination of synthetic and natural repeat settings. (Jn46, z4Ji)
- Causal methods strengthen analysis beyond correlational approaches. (Jn46, mGgi)

**Weaknesses:**

- Scope limited to simple repeats; generalization to complex tasks or multi-copy repeats unclear. (pHHB, z4Ji)
- Validation across broader PLM families (autoregressive, diffusion-based) needed. (Jn46)
- BLOSUM62-based counterfactuals may introduce bias. (Jn46)
- Presentation issues (confusing figures, inconsistent styles) affect clarity. (mGgi)

While the lack of validation across broader PLM families and the limited analysis of failure modes reduce the clarity of the contributions, the foundational ideas still hold significant promise. I have therefore decided to accept this paper.

**Additional Comments on Reviewer Discussion:**

- **Presentation Issues (Reviewer mGgi):** Confusing figures, inconsistent plot styles, and main-text/appendix discrepancies were noted. The authors committed to improvements in revision.

Overall, the authors' rebuttal addressed majority of the reviewers' concerns.  I suggest to accept this paper.